# MonoUNI: A Unified Vehicle and Infrastructure-side Monocular 3D Object Detection Network with Sufficient Depth Clues

**Jinrang Jia**[*]
Baidu Inc.
Beijing, China
jiajinrang@baidu.com

**Zhenjia Li**[*]
Baidu Inc.
Beijing, China
lizhenjia@baidu.com

**Yifeng Shi**[†]
Baidu Inc.
Beijing, China
shiyifeng@baidu.com

## Abstract

Monocular 3D detection of vehicle and infrastructure sides are two important topics in autonomous driving. Due to diverse sensor installations and focal lengths, researchers are faced with the challenge of constructing algorithms for the two topics based on different prior knowledge. In this paper, by taking into account the diversity of pitch angles and focal lengths, we propose a unified optimization target named normalized depth, which realizes the unification of 3D detection problems for the two sides. Furthermore, to enhance the accuracy of monocular 3D detection, 3D normalized cube depth of obstacle is developed to promote the learning of depth information. We posit that the richness of depth clues is a pivotal factor impacting the detection performance on both the vehicle and infrastructure sides. A richer set of depth clues facilitates the model to learn better spatial knowledge, and the 3D normalized cube depth offers sufficient depth clues. Extensive experiments demonstrate the effectiveness of our approach. Without introducing any extra information, our method, named MonoUNI, achieves state-of-the-art performance on five widely used monocular 3D detection benchmarks, including Rope3D and DAIR-V2X-I for the infrastructure side, KITTI and Waymo for the vehicle side, and nuScenes for the cross-dataset evaluation.

## 1  Introduction

Accurate 3D detection [14, 21, 25] is crucial for autonomous driving. Although LIDAR sensors [8, 18, 47, 51] provide high precision, camera sensors [7, 39, 52] are cost-effective and have a wider range of perception. Typically, autonomous driving systems employ frontal-view cameras mounted on the vehicle for 3D detection. As intelligent transportation continues to advance, there is increasing interest in using infrastructure-side cameras for 3D detection [13, 49, 53].

Due to the different installations and focal lengths between the vehicle and infrastructure-side cameras, researchers usually design algorithms to solve these two problems separately, which adds an additional application limitation. Fig. 1 illustrates the imaging process and the corresponding visual feature under different installations and focal lengths. With respect to the different installations, most vehicle-side cameras are installed on the top of the vehicle with a near-zero pitch angle, leading to a prior assumption that the optical axis is parallel to the ground. Most vehicle-side methods [4, 19, 27] are designed based on this assumption. In contrast, the infrastructure-side cameras, which are mounted on poles, typically have large pitch angles, rendering most of the existing vehicle-side

---

[*]Equal contribution.
[†]Corresponding author.

37th Conference on Neural Information Processing Systems (NeurIPS 2023).

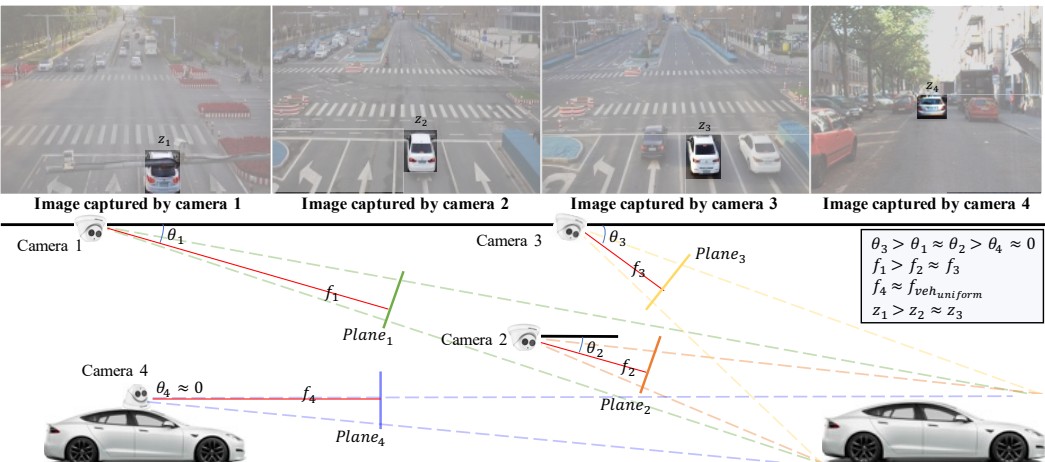

Figure 1: **Schematic diagram of the camera imaging process and visual feature under different focal lengths and pitch angles.** We assume that there is a scene where four cameras in different positions capture the same car (the white car highlighted in the four corresponding images). Camera 1,2,3 are infrastructure-side cameras, and camera 4 is the vehicle-side camera. The pitch angles of camera 1 and camera 2 are similar. Camera 2 is closer to the vehicle, and the focal length is relatively small. Due to the substantial variation in focal length between the two cameras, although the distance between the two cameras and the white car is significantly different, the visual features of the vehicle in the two images are nearly identical, which creates ambiguity in the depth estimation. The focal lengths of camera 2 and camera 3 are comparable. However, due to the different pitch angles of the two cameras, the same car at a similar depth appears with different visual features, resulting in increased difficulty for accurate depth regression. Camera 4 is on the vehicle side with a near-zero pitch angle, leading to a prior assumption that the optical axis is parallel to the ground which is not satisfied on the infrastructure side.

methods unsuitable for direct application, and the diversity of pitch angles also further increases the difficulty of object detection. According to the various focal lengths, the type of vehicle-side cameras are relatively uniform, and the focal lengths are thus similar, while the infrastructure-side cameras have a wide range of focal lengths. This results in a new challenge of 1-to-N ambiguity for depth estimation in monocular 3D detection, which has already been an ill-posed problem.

In this work, to address the above problems, we developed the obstacle projection models of the vehicle and infrastructure sides, and found that the former can actually be regarded as a special case of the latter, where the pitch angle is approximately 0 and the focal length is fixed. Consequently, we propose a unified optimization target: normalized depth, which is independent of focal length and pitch angle, so that the prediction of the obstacle depth on the two sides is no longer disturbed by the diversity of focal length and pitch angle. Furthermore, to enhance the performance of 3D detection, we draw inspiration from methods like AutoShape [26] and DID-M3D [37], which introduce new depth clues. For instance, AutoShape uses dense key points on the vehicle surface to establish geometric constraints, while DID-M3D employs depth maps to enable the network to predict object surface depth. They both add spatially correlated depth clues to the model in different ways with auxiliary data. We suggest that incorporating rich depth clues has a greater impact on depth estimation accuracy. Thus, we use the geometric relationship to create a 3D normalized cube depth for the obstacle, allowing the model to predict the normalized depth of the obstacle's 3D bounding box without introducing any extra data. By predicting the bias depth from each point on the 3D bounding box to the obstacle center, we can obtain the final depth conveniently. The approach of predicting obstacle 3D normalized cube depth maximizes depth clues and improves detection performance without additional data.

In summary, incorporating the above techniques, our method MonoUNI first achieves state-of-the-art (SOTA) performance on both vehicle and infrastructure-side monocular 3D detection. The main contributions of this work are as follows: 1) We proposed a unified optimization target, namely normalized depth, to address the differences between vehicle and infrastructure-side 3D detection caused by the diversity of pitch angle and focal length. This optimization target can be considered as

the standard solution for future monocular 3D detection of both sides. 2) We posit that rich depth clues are crucial in 3D detection, and thus propose the use of a 3D normalized cube depth to facilitate the learning of depth information. 3) Without using any additional information, MonoUNI ranks 1st in both the Rope3D [53] and DAIR-V2X-I [55] infrastructure-side benchmarks. Moreover, we apply our method to the vehicle-side benchmarks, such as KITTI [10] and Waymo [42], and also achieve competitive results. Cross-dataset evaluation of the KITTI val model on the nuScenes [3] val set demonstrates the generalizability of our method. The code is available at https://github.com/Traffic-X/MonoUNI.

## 2 Related Work

**Vehicle-side Monocular 3D Object Detection.**    Vehicle-side monocular 3D detection refers to the process of analyzing a single image captured by a camera mounted on a vehicle to predict the 3D locations, dimensions, and orientations of the interest obstacles. These methods can be mainly categorized into two groups based on whether additional data is utilized. The first type of method uses additional data, such as depth maps [9, 28, 38], CAD models [5, 26, 35, 34] or LIDAR [6, 39] to enhance the detection accuracy. Pseudo-LIDAR methods [29, 46] utilize the depth map predicted by the additional network to assist the monocular image to generate a pseudo point cloud and then adopt existing LIDAR-based 3D object detection pipeline. DID-M3D [37] uses a dense depth map to generate visual depth for powerful data augmentation. CaDDN [39] uses LIDAR points to supervise additional monocular network estimates dense depth map and converts the feature to BEV perspective for prediction. AutoShape [26] utilizes CAD models to generate dense key points to alleviate the sparse constraints. NeurOCS [32] attempts to introduce the Neural Radiance Field (NeRF) [31] to solve the problem of lack of supervision information. Mix-Teaching [50] utilizes unlabeled data to achieve semi-supervised learning. Although utilizing additional information allows these methods to achieve improved performance, it inevitably leads to increased labeling and computational costs. Moreover, obtaining such data is challenging in various scenarios, especially when it comes to the infrastructure side.

The second kind of method only uses a single image as input without any extra information. Some methods [23, 27, 45, 56] use geometric projection assumptions to improve the accuracy of 3D detection. GUPNet [27] uses the 2D and 3D heights of the object to construct similar triangles to assist in regression depth. MonoFlex [56] and MonoDDE [23] further extend this similar triangle relationship using the position of key corner points to assist regression depth. PGD [45] constructs geometric relation graphs across predicted objects and uses the graph to facilitate depth estimation. Due to the installed pitch angle of infrastructure-side cameras, these geometric projections cannot be applied to infrastructure-side monocular 3D detection. Other methods [25, 30, 44] directly predict the dimensions, orientations, and locations of obstacles without the aid of geometric projections. However, these methods are designed for the vehicle side, ignoring the pitch angle diversity and the ambiguity introduced by the focal length gap between different cameras on the infrastructure side.

**Infrastructure-side Monocular 3D Object Detection.**    Compared to vehicle-side monocular 3D object detection, which is typically limited to short-range perception, infrastructure-side 3D detection can overcome the problem of frequent occlusion in vehicle-based detection by increasing the sensor installation height, thereby providing long-range perception capabilities. Recently, DAIR-V2X-I [55] and Rope3D [53] are proposed to promote the development of 3D perception in infrastructure-side scenarios. Nonetheless, the challenges mentioned above, including the focal length difference and the variable pitch angle of the cameras, make it arduous to transfer the vehicle-side 3D detection method to the infrastructure-side setting, which results in relatively sluggish progress in infrastructure-side monocular 3D detection. BEVHeight [49] mitigates the issues arising from variations in camera pose parameters by directly predicting object height instead of object depth and achieves competitive performance in Rope3D and DAIR-V2X-I.

## 3 Method

### 3.1 Overview

Monocular 3D object detection extracts features from a single RGB image, predicting the category and 3D bounding box for each object in the image. The 3D bounding box can be further divided

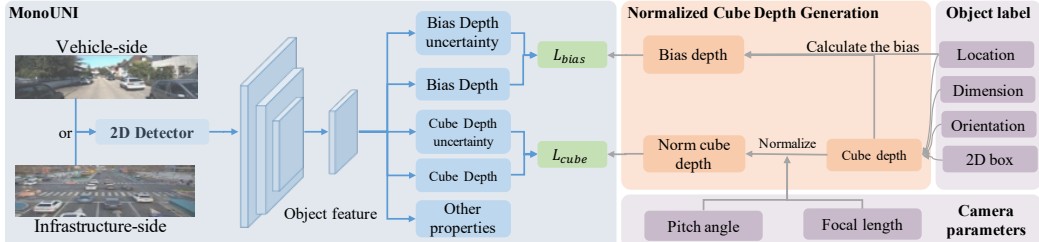

Figure 2: **Overview.** The left side of the picture describes the network architecture of MonoUNI, which uses a 2D detector to obtain object features, and then adopts different heads to estimate the cube depth, bias depth, and corresponding uncertainty, as well as other 3D properties. The right side depicts how we use the 3D label to generate the normalized cube depth and bias depth.

into 3D center location $(x, y, z)$, dimension $(h, w, l)$ and orientation (yaw angle) $\theta$. Among these properties, dimension, and orientation are easily learned by the network due to their strong correlation with visual features [15], but the 3D location is challenging because depth estimation is ill-posed.

The main idea of MonoUNI is to unify 3D detection targets for both vehicle and infrastructure sides, and further achieve accurate 3D location. The overall framework is depicted in Fig. 2. We use CenterNet [57] as the base model to generate discriminative representation, with DLA34 [54] serving as the backbone for feature extraction from images. Several network heads are established in MonoUNI to predict object properties, such as categorical heatmap, 2D bounding box, 3D offset, dimension, orientation, 3D normalized cube depth, bias depth, and depth uncertainty items [16, 27]. Depth uncertainty is widely used in 3D detection, which can enhance the loss's robustness against noisy inputs. The loss function is roughly similar to DID-M3D [37]. In the inference process, by leveraging the predicted 3D normalized cube depth and bias depth, and utilizing the known pitch angle and focal length information, our method can obtain the final obstacle depth.

## 3.2 Unified Optimization Target

**Problem Analysis.** In order to construct a unified optimization target for the vehicle and infrastructure sides, we analyze the significant difference in monocular 3D detection between them. First, the camera on the infrastructure side is typically installed at an elevated position with a specific pitch angle (the angle between the optical axis of the camera and the ground) to capture a wider view and detect more potential obstacles, which invalidates the commonly used assumption on the vehicle side that the optical axis of the camera is parallel to the ground. Not only that, due to different scenarios and installation methods, the pitch angle of the installed camera is varied. For instance, in the Rope3D dataset, the pitch angle of the camera ranges from 5 to 20 degrees. Second, to cope with different situations, the internal parameters of different infrastructure-side cameras usually have a relatively large difference. For example, the camera focal length of Rope3D ranges from 2100 to 2800 pixels, while the camera focal lengths of the KITTI dataset are all between 715 pixels and 721 pixels. The huge gap in focal length can lead to an ambiguity that obstacles of similar visual features in two images captured by cameras with different focal lengths can have different depths. To explore the influence of focal length diversity, we conducted experiments on Rope3D using both the popular single-stage and two-stage monocular object detection methods. Since the focal length of most images is centered around 2100 and 2700 pixels, we further split the dataset into train_2100, train_2700, val_2100, and val_2700. The experimental results are shown in Table 1.

As a consequence of the aforementioned ambiguity, utilizing two train sets for mixed training will decrease the accuracy of both validation sets when compared to training the network solely with images in a single focal length range. In particular, the average precision (AP) of GUPNet experiences a significant reduction of 70% on the val_2700. This decrease in accuracy can be attributed to the dominance of images with a focal length of around 2100 pixels during the training process.

**Normalized Depth.** In this subsection, we first build the simple projection models of the vehicle (Fig. 3 (a)) and infrastructure sides (Fig. 3 (b)). Taking the infrastructure-side projection model as an example, $O$ in the figure is the optical center of the camera, the ray $OZ$ denotes the optical axis (z-axis), point $C$ represents the center point of the obstacle, $P$ represents the intersection point

Table 1: **Analysis for different focal lengths on Rope3D dataset with new train/val division.**

| Method | Train_set | $AP_{3D}(IOU = 0.5\|R_{40})$ | | |
|---|---|---|---|---|
| | | val_2100 | val_2700 | val_all |
| GUPNet [27] | train_2100 | 13.20 | 0.03 | 7.42 |
| GUPNet [27] | train_2700 | 0.17 | 21.65 | 3.07 |
| GUPNet [27] | train_all | 10.82 | 5.85 | 9.38 |
| SMOKE [25] | train_2100 | 9.77 | 0.13 | 6.19 |
| SMOKE [25] | train_2700 | 0.04 | 23.20 | 3.64 |
| SMOKE [25] | train_all | 6.04 | 18.01 | 8.48 |

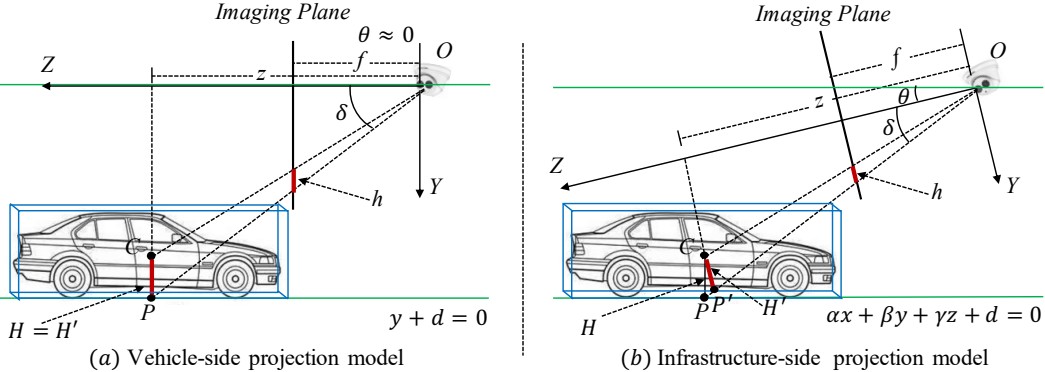

(a) Vehicle-side projection model    (b) Infrastructure-side projection model

Figure 3: **Simple projection models of the vehicle and infrastructure-side.**

between the vertical line from the center point $C$ to the ground and the ground plane, $H$ denotes the 3D distance of $PC$, and $h$ is the pixel distance from the center point $C$ to the ground in the imaging plane, $\theta$ is the pitch angle of the camera, $z$ denotes the depth of center point $C$, $f$ represents the focal length and $\delta$ is the included angle between the line connecting the point $P$ to the optical center $O$ and $OZ$. Extend a line from the obstacle's center point $C$ along the camera's imaging plane direction, intersecting line $OP$ at point $P'$. The distance $CP'$ is denoted as $H'$. According to the parallel relationship, the following can be easily deduced:

$$\frac{H'}{h} = \frac{z}{f} \tag{1}$$

After a simple geometric calculation, the following equation can be derived[3]:

$$H' = H * (\cos\theta - \sin\theta * \tan\delta) \tag{2}$$

Substitute it into equation (1) and we get:

$$z = \frac{H}{h} * (\cos\theta - \sin\theta * \tan\delta) * f \tag{3}$$

Among them, $H$ and $h$ are easy to learn based on visual features, while $\delta$, $\theta$, and $f$ are challenging to directly learn from the image. Therefore, learning $z$ directly requires the model to have a strong ability to infer focal length and angle information. However, from the above analysis, it is difficult or even ambiguous. Typically, the focal length $f$ can be obtained as prior knowledge, the pitch angle $\theta$ can be calculated as $\theta = \arctan(\gamma/\beta)$ from the ground equation $\alpha x + \beta y + \gamma z + d = 0$, and $\delta$ can be calculated through simple calculation $\delta = \arctan((v_p - c_y)/f)$, where $v_p$ is the y-axis pixel coordinates of Point $P$ and $c_y$ is the principal point in y-axis. In particular, $v_p$ can be simply and roughly replaced by $v_c$ which may introduce an average relative error of 2.9% (statistics based on the

---

[3]See the Supplementary Material for more proof details.

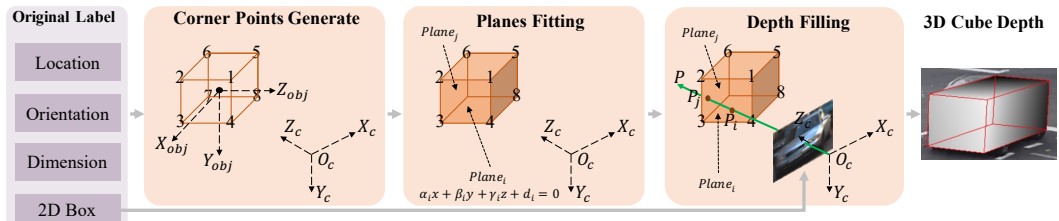

Figure 4: **The generation process of the 3D cube depth.**

Rope3D dataset), or be predicted by the model. Therefore, a simple transformation of the formula (3) yields:

$$Normalized\_depth = \frac{z}{(\cos\theta - \sin\theta * \tan\delta) * f} \tag{4}$$

Formula (4) represents the new unified optimization target: normalized depth, which makes depth prediction independent of pitch angle and focal length, simplifying the task difficulty. When $\theta$ is equal to 0, the normalized depth degenerates into a situation where only the focal length is used for normalization ($\frac{z}{f}$), that is, the vehicle-side projection model. In fact, simplifying the normalized depth with either the focal length ($\frac{z}{f}$) or pitch angle ($\frac{z}{\cos\theta - \sin\theta * \tan\delta}$) can also partially reduces the learning difficulty and improve detection accuracy, which we will analyze in detail in the ablation experiments in section 4.4. During inference, the model outputs normalized depth, and we multiply it by the denominator in the formula (4) to get the final instance depth.

### 3.3   3D Normalized Cube Depth

To enrich depth clues and further improve detection performance, we utilize original labels to generate 3D cube depth for each obstacle. Based on the analysis in section 3.2, our 3D cube depth need to be normalized and become 3D normalized cube depth finally. The generation process of the 3D cube depth is shown in Fig. 4. Firstly, we use the location, orientation, and dimension information from the label to obtain the positions of the 8 corner points in the 3D bounding box in the camera coordinate system. These points are then marked in a specific sequence (1-8). Secondly, we calculate the plane equation $\alpha_i x + \beta_i y + \gamma_i z + d_i = 0$ for each surface of the obstacle. To determine the equation, we select any 3 of the 4 vertices of the surface since a plane can be uniquely defined by 3 non-collinear points. Thirdly, we generate the 3D cube depth for the obstacle and project it onto the imaging plane. Specifically, each pixel $(u, v)$ in the 2D bounding box is traversed to determine whether it belongs to any surface in the 2D image. Then, utilizing the internal parameters of the camera, the intersection point $(x, y, z)$ of the pixel's ray with the corresponding plane can be computed and the depth $z$ is thus achieved. The mathematical expression for this computation is as follows:

$$\begin{cases} \alpha_i x + \beta_i y + \gamma_i z + d_i = 0, for\ i = 1,...,6 \\ f_x \frac{x}{z} + c_x = u \\ f_y \frac{y}{z} + c_y = v \end{cases} \tag{5}$$

where $f_x$ and $f_y$ are the focal length of the camera, and $c_x$ and $c_y$ are the principal points. $i$ denotes the index of the surface on the 3D bounding box. The depth $z$ can be calculated as:

$$z = \frac{-d_i}{\frac{\alpha_i(u-c_x)}{f_x} + \frac{\beta_i(v-c_y)}{f_y} + \gamma_i} \tag{6}$$

In cases where a pixel belongs to multiple faces in the 2D image, we select the smallest depth value as the final depth. This is because, during the camera imaging process, points with larger depth values are obstructed by ones closer to the camera. The same strategy is also applied when a pixel simultaneously belongs to multiple obstacles. Once the 3D cube depth is obtained, we apply Equation 3 to each depth value in the cube, resulting in a 3D normalized cube depth that can be used for

Table 2: **Monocular 3D detection performance of Car category on Rope3D** *val*, **DAIR-V2X-I** *val* **and KITTI** *test* **sets.** We highlight the best results in bold and the second ones in underlined. For the extra data, the first column means whether using depth maps as additional input on the infrastructure side and the second column means on the vehicle side. '-' means that no official results or reasonable reproduction results in the new dataset.

| Method | Extra Data | Rope3D | | DAIR | | | KITTI | | |
|---|---|---|---|---|---|---|---|---|---|
| | | $AP_{3D}$ | $R_{score}$ | Easy | Mod. | Hard | Easy | Mod. | Hard |
| M3D-RPN [1] | Depth \| None | 67.17 | 73.14 | - | - | - | 14.76 | 9.71 | 7.42 |
| MonoDLE [30] | Depth \| None | 77.50 | 80.84 | - | - | - | 7.23 | 12.26 | 10.29 |
| MonoFlex [56] | Depth \| None | 59.78 | 66.66 | - | - | - | 19.94 | 13.89 | 12.07 |
| DID-M3D [37] | None \| Depth | - | - | - | - | - | 24.40 | 16.29 | 13.75 |
| CMKD [11] | None \| Depth | - | - | - | - | - | 25.09 | 16.99 | 15.30 |
| LPCG+MonoFlex [36] | None \| Depth | - | - | - | - | - | 25.56 | 17.80 | 15.38 |
| MonoEF [58] | None \| None | - | - | - | - | - | 21.29 | 13.87 | 11.71 |
| DEVIANT [20] | None \| None | - | - | - | - | - | 21.88 | 14.46 | 11.89 |
| MonoCon [48] | None \| None | - | - | - | - | - | 22.50 | 16.46 | 13.95 |
| MonoATT [60] | None \| None | - | - | - | - | - | 24.72 | 17.37 | 15.00 |
| MoGDE [59] | None \| None | - | - | - | - | - | **27.07** | **17.88** | **15.66** |
| Kinematic3D [2] | None \| None | 50.57 | 58.86 | - | - | - | 19.07 | 12.72 | 9.17 |
| SMOKE [25] | None \| None | 72.13 | 76.26 | 66.03 | 62.24 | 60.71 | 14.03 | 9.76 | 7.84 |
| GUPNet [27] | None \| None | 66.52 | 70.14 | 62.22 | 55.94 | 55.90 | 22.26 | 15.02 | 13.12 |
| Imvoxelnet [40] | None \| None | - | - | 44.78 | 37.58 | 37.55 | 17.15 | 10.97 | 9.15 |
| BEVFormer [24] | None \| None | 50.62 | 58.78 | 61.37 | 50.73 | 50.73 | - | - | - |
| BEVDepth [22] | None \| None | 69.63 | 74.70 | 75.50 | 63.58 | 63.67 | - | - | - |
| BEVHeight [49] | None \| None | 74.60 | 78.72 | 77.78 | 65.77 | 65.85 | - | - | - |
| **MonoUNI(Ours)** | None \| None | **92.45** | **92.63** | **90.92** | **87.24** | **87.20** | 24.75 | 16.73 | 13.49 |

supervision. In order to get the center point depth, we also supervise the bias depth from each point on the cube depth to the center point depth. The utilization of cube depth can maximize the enrichment of depth clues based on existing labels without additional information such as depth maps, CAD, or LIDAR points. Compared with only supervising the depth of the center point and corner points, the 3D cube depth is a sufficient way to utilize the depth information.

# 4 Experiments

## 4.1 Dataset and Evaluation Metrics

**Rope3D.** Rope3D [53] is a comprehensive real-world benchmark for infrastructure-side 3D detection, featuring over 50,000 images and more than 1.5 million 3D objects in diverse scenes. The benchmark comprises various settings, such as different cameras with ambiguous mounting positions, diverse camera specifications, as well as different environmental conditions. For the evaluation metrics, we follow the official settings using $AP_{3D|R40}$ and $Rope_{score}$ [53], which is a combined metric of the 3D AP and other similarities, such as Average Ground Center Similarity (ACS). We follow the proposed homologous setting to utilize 70% of the images as training, and the remaining as validating. All images are randomly sampled.

**DAIR-V2X-I.** DAIR-V2X-I [55] is a subset of DAIR-V2X, which is a large-scale multimodal vehicle-infrastructure collaborative perception dataset. It contains 10,000 infrastructure-side images and 493,000 3D bounding boxes with 10 classes. We follow the official train/val division for evaluation, and use $AP_{3D|R40}$ as the main metric, with an IOU threshold of 0.5.

**KITTI.** KITTI [10] is a widely used benchmark consisting of 7481 training images and 7518 testing images for vehicle-side monocular 3D detection. The dataset contains three classes: Car, Pedestrian, and Cyclist, each with three difficulty levels: Easy, Moderate, and Hard. Moderate difficulty is considered the official rank level of the KITTI leaderboard. To ensure a fair comparison with previous methods, results are submitted to an official server for evaluation on the test set. $AP_{3D|R40}$ is used as the main metric with IOU thresholds of 0.7 for cars, and 0.5 for Pedestrians and Cyclists respectively.

Table 3: **Monocular 3D detection performance of Pedestrian and Cyclist category on KITTI *test* set.**

| Method | $AP_{3D}$ | | | | | |
|---|---|---|---|---|---|---|
| | Pedestrian | | | Cyclist | | |
| | Easy | Mod. | Hard | Easy | Mod. | Hard |
| DFR-Net [61] | 6.09 | 3.62 | 3.39 | 5.69 | 3.58 | 3.10 |
| CaDDN [39] | 12.87 | 8.14 | 6.76 | 7.00 | 3.41 | 3.30 |
| MonoCon [48] | 13.10 | 8.41 | 6.94 | 2.80 | 1.92 | 1.55 |
| GUPNet [27] | 14.95 | 9.76 | 8.41 | 5.58 | 3.21 | 2.66 |
| MonoDTR [12] | 15.33 | 10.18 | 8.61 | 5.05 | 3.27 | 3.19 |
| **MonoUNI** | **15.78** | **10.34** | **8.74** | **7.34** | **4.28** | **3.78** |

Table 4: **Monocular 3D detection performance of Big Vehicle category on Rope3D *val* set.**

| Method | Big Vehicle | |
|---|---|---|
| | $AP_{3D}$ | $R_{score}$ |
| M3D-RPN [1] | 32.19 | 40.52 |
| MonoDLE [30] | 44.71 | 52.83 |
| SMOKE [25] | 52.04 | 59.23 |
| GUPNet [27] | 45.27 | 52.19 |
| BEVFormer [24] | 34.58 | 45.16 |
| BEVDepth [22] | 45.02 | 54.64 |
| BEVHeight [49] | 48.93 | 57.70 |
| **MonoUNI** | **76.30** | **79.20** |

Table 5: **Ablation Study on different components of our overall framework on Rope3D and KITTI *val* set for Car category**.

| Experiments | Normlized Depth | | Cube | Rope3D | | KITTI | | |
|---|---|---|---|---|---|---|---|---|
| | Focal | Pitch | | $AP_{3D}$ | $R_{score}$ | Easy | Mod. | Hard |
| (a) | | | | 79.37 | 81.38 | 21.71 | 14.93 | 12.10 |
| (b) | ✓ | | | 83.42 | 84.91 | 22.35 | 15.29 | 12.34 |
| (c) | | ✓ | | 81.55 | 83.53 | 21.68 | 14.85 | 12.12 |
| (d) | ✓ | ✓ | | 86.69 | 87.99 | 22.19 | 14.94 | 12.45 |
| (e) | | | ✓ | 87.07 | 88.16 | **24.66** | 17.07 | **14.06** |
| (f) | ✓ | ✓ | ✓ | **92.45** | **92.63** | 24.51 | **17.18** | 14.01 |

**Waymo.** Waymo [42] assesses objects using a dual-tiered categorization: Level 1 and Level 2. The assessment is performed across three distance intervals: [0, 30), [30, 50), and [50, ∞) meters. Waymo employs the $APH_{3D}$ percentage metric, which integrates heading data into the $AP_{3D}$, as a reference benchmark for evaluation.

**nuScenes.** nuScenes [3] comprises 28,130 training and 6,019 validation images captured from the front camera. We use validation split for cross-dataset evaluation.

## 4.2 Implementation Details

Our proposed MonoUNI is trained on 4 Tesla V100 GPUs with a batch size of 16 for 150 epochs. We use Adam [17] as our optimizer with an initial learning rate $1.25 \times e - 3$. Images are all resized to the same size of $960 \times 512$ for infrastructure side and $1280 \times 384$ for vehicle side. Following [37], the ROI-Align size $d \times d$ is set to $7 \times 7$. Inspired by [33], we adopt the multi-bin strategy for heading angle and depth prediction in our baseline. Random crop and expand along principal points are used to achieve more physical data augmentation.

## 4.3 Main Results

**Results of Car Category.** Since there was no previous method designed for supporting both the vehicle and infrastructure sides at the same time, the results of some methods were reproduced by us or other published papers (BEVHeight [49], Rope3D [53] and DAIR-V2X [55]). As shown in Table 2, our proposed MonoUNI achieves superior performance than previous methods on infrastructure-side benchmarks, even those with extra data. Specifically, compared with BEVHeight which is the recent top1-ranked image-only method for infrastructure side, MonoUNI gains significant improvement of **17.85%/13.91%** in $AP_{3D}$ and $R_{score}$ on Rope3D benchmark. According to DAIR-V2X, our method achieves over **21%** improvement compared with BEVHeight on Moderate difficulty. For vehicle-side benchmark, although our method did not rank first, it reached a competitive result on Easy difficulty. On the Moderate and Hard difficulties, MonoUNI is slightly inferior to methods such as MonoATT [60] and MoGDE [59], probably because the gain of the 3D cube depth is weakened in the case of far-distance or severe truncation.

**Results of Other Categories.** In Table 3, we present the results of pedestrians and cyclists on the test set of KITTI. MonoUNI outperforms all methods by a large margin. This may be because the

Table 6: **Analysis of MonoUNI for different focal lengths on Rope3D with new *train/val* division.**

| Method | Train | $AP_{3D}(IOU = 0.5\|R_{40})$ val_2100 | val_2700 | val_all |
|---|---|---|---|---|
| MonoUNI | 2100 | 25.78 | 21.30 | 24.47 |
| MonoUNI | 2700 | 5.77 | 23.42 | 9.25 |
| MonoUNI | all | 26.63 | 38.10 | 28.91 |

Table 7: **Cross-dataset evaluation of the KITTI *val* model on KITTI *val* and nuScenes *val* cars with depth MAE.**

| Method | KITTI Val | | | | nuScenes frontal Val | | | |
|---|---|---|---|---|---|---|---|---|
| | 0-20 | 20-40 | 40-∞ | All | 0-20 | 20-40 | 40-∞ | All |
| M3D-RPN [1] | 0.56 | 1.33 | 2.73 | 1.26 | 0.94 | 3.06 | 10.36 | 2.67 |
| MonoRCNN [41] | 0.46 | 1.27 | 2.59 | 1.14 | 0.94 | 2.84 | 8.65 | 2.39 |
| GUPNet [27] | 0.45 | 1.10 | 1.85 | 0.89 | 0.82 | _1.70_ | 6.20 | 1.45 |
| DEVIANT [20] | 0.40 | 1.09 | 1.80 | 0.87 | _0.76_ | **1.60** | **4.50** | **1.26** |
| **MonoUNI** | **0.38** | **0.92** | **1.79** | **0.865** | **0.72** | 1.79 | _4.98_ | _1.43_ |

Table 8: **Monocular 3D detection performance of Vehicle category on Waymo *val* set.**

| $IOU_{3D}$ | Difficulty | Method | Extra | $AP_{3D}$ All | 0-30 | 30-50 | 50-∞ | $APH_{3D}$ All | 0-30 | 30-50 | 50-∞ |
|---|---|---|---|---|---|---|---|---|---|---|---|
| 0.7 | Level_1 | CaDDN [39] | LIDAR | 5.03 | 15.54 | 1.47 | 0.10 | 4.99 | 14.43 | 1.45 | 0.10 |
| | | PatchNet [28] in [43] | Depth | 0.39 | 1.67 | 0.13 | 0.03 | 0.39 | 1.63 | 0.12 | 0.03 |
| | | PCT [43] | Depth | 0.89 | 3.18 | 0.27 | 0.07 | 0.88 | 3.15 | 0.27 | 0.07 |
| | | M3D-RPN [1] in [39] | None | 0.35 | 1.12 | 0.18 | 0.02 | 0.34 | 1.10 | 0.18 | 0.02 |
| | | GUPNet [27] in [20] | None | 2.28 | 6.15 | 0.81 | _0.03_ | 2.27 | 6.11 | 0.80 | _0.03_ |
| | | DEVIANT [20] | None | _2.69_ | _6.95_ | **0.99** | 0.02 | _2.67_ | _6.90_ | **0.98** | 0.02 |
| | | **MonoUNI (Ours)** | None | **3.20** | **8.61** | _0.87_ | **0.13** | **3.16** | **8.50** | _0.86_ | **0.12** |
| 0.7 | Level_2 | CaDDN [39] | LIDAR | 4.49 | 14.50 | 1.42 | 0.09 | 4.45 | 14.38 | 1.41 | 0.09 |
| | | PatchNet [28] in [43] | Depth | 0.38 | 1.67 | 0.13 | 0.03 | 0.36 | 1.63 | 0.11 | 0.03 |
| | | PCT [43] | Depth | 0.66 | 3.18 | 0.27 | 0.07 | 0.66 | 3.15 | 0.26 | 0.07 |
| | | M3D-RPN [1] in [39] | None | 0.35 | 1.12 | 0.18 | 0.02 | 0.33 | 1.10 | 0.17 | 0.02 |
| | | GUPNet [27] in [20] | None | 2.14 | 6.13 | 0.78 | 0.02 | 2.12 | 6.08 | 0.77 | 0.02 |
| | | DEVIANT [20] | None | _2.52_ | _6.93_ | **0.95** | _0.02_ | _2.50_ | _6.87_ | **0.94** | _0.02_ |
| | | **MonoUNI (Ours)** | None | **3.04** | **8.59** | _0.85_ | **0.12** | **3.00** | **8.48** | _0.84_ | **0.12** |
| 0.5 | Level_1 | CaDDN [39] | LIDAR | 17.54 | 45.00 | 9.24 | 0.64 | 17.31 | 44.46 | 9.11 | 0.62 |
| | | PatchNet [28] in [43] | Depth | 2.92 | 10.03 | 1.09 | 0.23 | 2.74 | 9.75 | 0.96 | 0.18 |
| | | PCT [43] | Depth | 4.20 | 14.70 | 1.78 | 0.39 | 4.15 | 14.54 | 1.75 | 0.39 |
| | | M3D-RPN [1] in [39] | None | 3.79 | 11.14 | 2.16 | _0.26_ | 3.63 | 10.70 | 2.09 | 0.21 |
| | | GUPNet [27] in [20] | None | 10.02 | 24.78 | 4.84 | 0.22 | 9.94 | 24.59 | _4.78_ | _0.22_ |
| | | DEVIANT [20] | None | _10.98_ | **26.85** | **5.13** | 0.18 | **10.89** | **26.64** | **5.08** | 0.18 |
| | | **MonoUNI (Ours)** | None | **10.98** | _26.63_ | 4.04 | **0.57** | _10.73_ | _26.30_ | 3.98 | **0.55** |
| 0.5 | Level_2 | CaDDN [39] | LIDAR | 16.51 | 44.87 | 8.99 | 0.58 | 16.28 | 44.33 | 8.86 | 0.55 |
| | | PatchNet [28] in [43] | Depth | 2.42 | 10.01 | 1.07 | 0.22 | 2.28 | 9.73 | 0.97 | 0.16 |
| | | PCT [43] | Depth | 4.03 | 14.67 | 1.74 | 0.36 | 4.15 | 14.51 | 1.71 | 0.35 |
| | | M3D-RPN [1] in [39] | None | 3.61 | 11.12 | 2.12 | _0.24_ | 3.46 | 10.67 | 2.04 | _0.20_ |
| | | GUPNet [27] in [20] | None | 9.39 | 24.69 | _4.67_ | 0.19 | 9.31 | 24.50 | _4.62_ | 0.19 |
| | | DEVIANT [20] | None | _10.29_ | **26.75** | **4.95** | 0.16 | _10.20_ | **26.54** | **4.90** | 0.16 |
| | | **MonoUNI (Ours)** | None | **10.38** | _26.57_ | 3.95 | **0.53** | **10.24** | _26.24_ | 3.89 | **0.51** |

introduction of depth clues promotes pedestrians and cyclists better learn spatial features. Table 4 shows the results of Big Vehicle on the Rope3D benchmark.

## 4.4 Ablation Study

**Effectiveness of Different Components in MonoUNI on Rope3D and KITTI *val* set for Car category.** We evaluate the effectiveness of the normalized depth and the 3D cube depth through ablations on two datasets. As shown in Table 5, the normalized depth, particularly the focal length, has a significant improvement on the infrastructure side, but only a minor effect on the vehicle side. This is due to the fact that the focal length of the camera is similar across the KITTI data, and the pitch angle is close to 0. On the other hand, the cube depth improves results for both the vehicle and infrastructure sides, providing evidence of its efficacy.

**Analysis for different focal lengths.** Table 6 presents the performance of MonoUNI at different focal lengths. Compared with the results in Table 1, our method has obvious benefits in solving the problem of focal length diversity. In the case of using all training data, our method can improve 19.53% and 20.43% AP respectively compared with GUPNet and SMOKE.

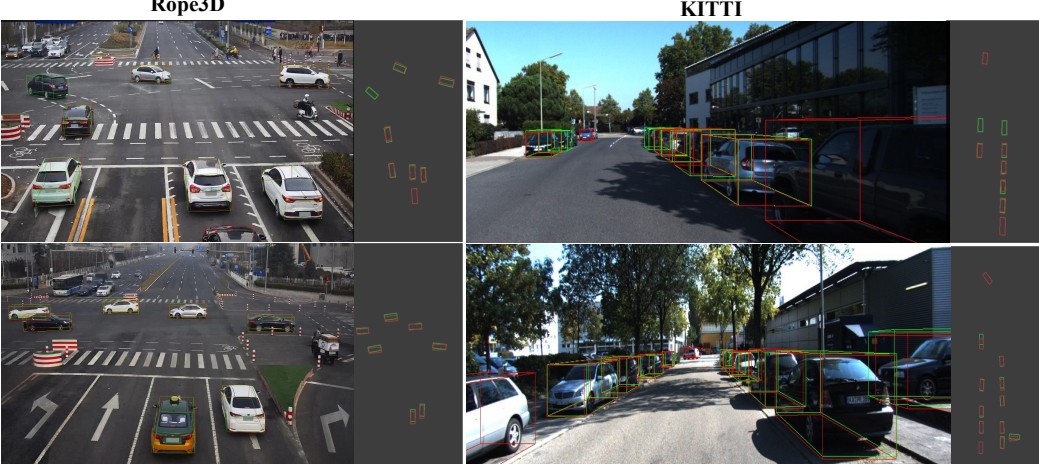

Figure 5: **Qualitative visualization on the Rope3D and KITTI val set.** The 3D green boxes are produced by MonoUNI and the red boxes are the ground truths.

**Cross-dataset evaluation.**    Tabel 7 shows the result of our KITTI val model on the KITTI val and nuScenes [3] frontal val images, using mean absolute error (MAE) of the depth [41]. MonoUNI is better than GUPNet [27] and achieves similar competitive performance to DEVIANT [20]. This is because DEVIANT is equivariant to the depth translations and is more robust to data distribution changes.

**Performance of cube depth on large dataset Waymo.**    In order to fully verify the effectiveness of cube depth, we verified MonoUNI on Waymo, which is a large-scale vehicle-side 3D detection dataset, as shown in Table 8.

### 4.5   Qualitative Results

We visualize the detection results of the MonoUNI on the both vehicle and infrastructure sides in Fig. 5. As can be seen, the MonoUNI can accurately estimate the 3D position of objects, even those not labeled by the annotators due to severe occlusion in the first row of KITTI. However, for far-distance and severely truncated obstacles, our model suffers from missed detections.

## 5   Conclusion

In this paper, we propose a new optimization target named normalized depth to unify monocular 3D detection for both vehicle and infrastructure sides, addressing the problem of focal length and pitch angle diversity. Furthermore, we introduce 3D Cube Depth as an additional supervision clue to improve the 3D detection performance. Experiments on five datasets (Rope3D, DAIR-V2X-I, KITTI, Waymo and nuScenes) fully demonstrate the effectiveness of our method.

## 6   Limitations and Future Work

Although the normalized depth unifies the optimization target of the vehicle and infrastructure sides, separate training for the two sides is still necessary instead of direct hybrid training. In the future, we aim to enable one model file to support 3D detection for both sides which would hold significant value for industrial applications, such as the domain gap problem in the vehicle-infrastructure collaborative perception and fusion. In addition, global depth clues such as depth similarity (i.e., contrastive learning) or the global topological relationship can be further developed.

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
