# Supplementary Material

**Jinrang Jia**[*]
Baidu Inc.
Beijing, China
jiajinrang@baidu.com

**Zhenjia Li**[*]
Baidu Inc.
Beijing, China
lizhenjia@baidu.com

**Yifeng Shi**[†]
Baidu Inc.
Beijing, China
shiyifeng@baidu.com

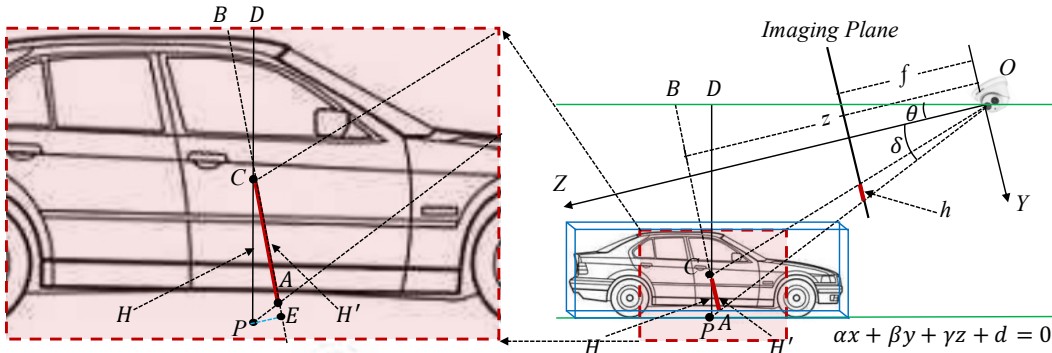

Figure 1: **Infrastructure-side projection model.** The left side is a zoom-in of the right side. As can be seen on the right side, $O$ in the figure is the optical center of the camera, the ray $OZ$ denotes the optical axis (z-axis), point $C$ represents the center point of the obstacle, $P$ represents the intersection point between the vertical line from the center point $C$ to the ground and the ground plane, $H$ denotes the 3D distance of $PC$, and $h$ is the pixel distance from the center point $C$ to the ground in the imaging plane, $\theta$ is the pitch angle of the camera, $z$ denotes the depth of center point $C$, $f$ represents the focal length and $\delta$ is the included angle between the line connecting the point $P$ to the optical center $O$ and $OZ$.

## 1 Detailed calculation process of the normalized depth

Our aim is to work out how to express the depth $z$ and further derive the normalized depth. First, since the image plane is parallel to the $AB$ ($AB$ is the virtual auxiliary plane set by us), we can get:

$$\frac{H'}{h} = \frac{z}{f} \tag{1}$$

where $H' = AC$. Here, $f$ is a known quantity and $h$ is easily obtained from visual features, so we only need to obtain $H'$. From point $P$ to straight line $AB$ draw an auxiliary vertical line and intersect $AB$ at point E, that is, $PE$ is perpendicular to $AB$. We can get:

$$H' = AC = CE - AE \tag{2}$$

Since:

$$CE = PC * \cos \angle ACP \tag{3}$$

---

[*]Equal contribution.
[†]Corresponding author.

37th Conference on Neural Information Processing Systems (NeurIPS 2023).

$$AE = PE * \tan \angle APE = PC * \sin \angle ACP * \tan \angle APE \quad (4)$$

$$H = PC \quad (5)$$

We have:

$$H' = H \cos \angle ACP - H \sin \angle ACP * \tan \angle APE \quad (6)$$

According to the parallel relationship, we can get: $\angle ACP = \theta$, $\angle APE = \delta$. Thus:

$$H' = H \cos \theta - H \sin \theta * \tan \delta = H * (\cos \theta - \sin \theta * \tan \delta) \quad (7)$$

Substitute it into equation (1) and we get:

$$z = \frac{H}{h} * (\cos \theta - \sin \theta * \tan \delta) * f \quad (8)$$

## 2 More detailed results on three benchmarks

### 2.1 Detailed results on KITTI

| Method | Reference | Extra Data | $AP_{3D}(IOU = 0.7|R_{40})$ | | | $AP_{BEV}(IOU = 0.7|R_{40})$ | | |
|--------|-----------|------------|------|------|------|------|------|------|
| | | | Easy | **Mod.** | Hard | Easy | Mod. | Hard |
| MonoPSR [7] | CVPR 2019 | LIDAR | 10.76 | 7.25 | 5.85 | 18.33 | 12.58 | 9.91 |
| PatchNet [14] | ECCV 2020 | Depth | 15.68 | 11.12 | 10.17 | 22.97 | 16.86 | 14.97 |
| D4LCN[5] | CVPR 2020 | Depth | 16.65 | 11.72 | 9.51 | 22.51 | 16.02 | 12.55 |
| MonoRUn [3] | CVPR 2021 | LIDAR | 19.65 | 12.30 | 10.58 | 27.94 | 17.34 | 15.24 |
| CaDDN [18] | CVPR 2021 | LIDAR | 19.17 | 13.41 | 11.46 | 27.94 | 18.91 | 17.19 |
| DFR-Net [29] | ICCV 2021 | Depth | 19.40 | 13.63 | 10.35 | 28.17 | 19.17 | 14.84 |
| AutoShape [12] | ICCV 2021 | CAD | 22.47 | 14.17 | 11.36 | 30.66 | 20.08 | 15.59 |
| DID-M3D [17] | ECCV 2022 | Depth | 24.40 | 16.29 | 13.75 | 32.95 | 22.76 | 19.83 |
| DD3D [16] | ICCV 2021 | Depth | 23.22 | 16.34 | 14.20 | 30.98 | 22.56 | 20.03 |
| MoGDE [27] | NeurIPS 2022 | Odometry | 27.07 | 17.88 | 15.66 | 38.38 | 25.60 | 22.91 |
| M3D-RPN [1] | ICCV 2019 | None | 14.76 | 9.71 | 7.42 | 21.02 | 13.67 | 10.23 |
| SMOKE [11] | CVPR 2020 | None | 14.03 | 9.76 | 7.84 | 20.83 | 14.49 | 12.75 |
| MonoPair [4] | CVPR2020 | None | 13.04 | 9.99 | 8.65 | 19.28 | 14.83 | 12.89 |
| MonoDLE [15] | CVPR2021 | None | 17.23 | 12.26 | 10.29 | 24.79 | 18.89 | 16.00 |
| PCT [21] | NeurIPS 2021 | None | 21.00 | 13.37 | 11.31 | 29.65 | 19.03 | 15.92 |
| MonoFlex [26] | CVPR 2021 | None | 19.94 | 13.89 | 12.07 | 28.23 | 19.75 | 16.89 |
| MonoEdge [28] | WACV 2023 | None | 21.08 | 14.47 | 12.73 | 28.80 | 20.35 | 17.57 |
| GUPNet [13] | ICCV 2021 | None | 22.26 | 15.02 | 13.12 | 30.29 | 21.19 | 18.20 |
| MonoDTR [6] | CVPR 2022 | None | 21.99 | 15.39 | 12.73 | 28.59 | 20.38 | 17.14 |
| MonoCon [22] | AAAI 2022 | None | 22.50 | 16.46 | 13.95 | 31.12 | 22.10 | 19.00 |
| **MonoUNI(Ours)** | - | None | 24.75 | 16.73 | 13.49 | 33.28 | 23.05 | 19.39 |

Table 1: **Monocular 3D detection performance of Car category on KITTI *test* set.** All results are evaluated on KITTI testing server. Same as KITTI leaderboard, methods are ranked under the moderate difficulty level. For the extra data: 1) **LIDAR** denotes methods use extra LIDAR cloud points in training process. 2) **Depth** means utilizing depth maps or models pre-trained under another depth estimation dataset. 3) **CAD** denotes using dense shape annotations provided by CAD models. 4) **Odometry** means utilizing extra odometry poses, images, or a well-trained network. 5) **None** means no extra data is used.

Table 1 lists the results of the vehicle-side 3D detection methods on the KITTI dataset in recent years. Our method ranks first among methods that do not use any additional data, which is improved $(2.25\%/0.27\%/-0.46\%)$ in $AP_{3D}$ and $(2.16\%/0.95\%/0.39\%)$ in $AP_{BEV}$ compared to MonoCon [22].

## 2.2 Detailed results on Rope3D

Table 2 shows the comparison of the results of methods on the Rope3D dataset in recent years. Compared with other methods, our method has greatly improved $AP$ and $Rope_{score}$ which ranks first.

| Method | Extra Data | $IOU = 0.5$ | | | | $IOU = 0.7$ | | | |
| | | Car | | Big Vehicle | | Car | | Big Vehicle | |
| | | $AP$ | $Rope$ | $AP$ | $Rope$ | $AP$ | $Rope$ | $AP$ | $Rope$ |
|---|---|---|---|---|---|---|---|---|---|
| M3D-RPN [1] | Ground | 54.19 | 62.65 | 33.05 | 44.94 | 16.75 | 32.90 | 6.86 | 24.19 |
| M3D-RPN [1] | Depth | 67.17 | 73.14 | 39.06 | 49.95 | 33.94 | 46.45 | 11.28 | 28.12 |
| Kinematic3D [2] | Ground | 50.57 | 58.86 | 37.60 | 48.08 | 17.74 | 32.99 | 6.10 | 22.88 |
| MonoDLE [15] | Ground | 51.70 | 60.36 | 40.34 | 50.07 | 13.58 | 29.46 | 9.63 | 25.80 |
| MonoDLE [15] | Depth | 77.50 | 80.84 | 49.07 | 57.22 | 54.53 | 62.48 | 17.25 | 32.00 |
| MonoFlex [26] | Ground | 60.33 | 66.86 | 37.33 | 47.96 | 33.78 | 46.12 | 10.08 | 26.16 |
| MonoFlex [26] | Depth | 59.78 | 66.66 | 59.81 | 66.07 | 35.64 | 47.43 | 24.61 | 38.01 |
| BEVFormer [10] | None | 50.62 | 58.78 | 34.58 | 45.16 | 24.64 | 38.71 | 10.05 | 25.56 |
| BEVDepth [9] | None | 69.63 | 74.70 | 45.02 | 54.64 | 42.56 | 53.05 | 21.47 | 35.82 |
| BEVHeight [24] | None | 74.60 | 78.72 | 48.93 | 57.70 | 45.73 | 55.62 | 23.07 | 37.04 |
| **MonoUNI(Ours)** | None | 92.45 | 92.63 | 76.30 | 79.20 | 74.50 | 78.26 | 43.04 | 52.63 |

$AP$ and $Rope$ denote $AP_{3D|R40}$ and $Rope_{score}$ respectively.

Table 2: **Monocular 3D detection performance of Car and Big Vehicle categories on Rope3D *val* set.**

## 2.3 Detailed results on DAIR-V2X-I

| Method | M | $Veh.$(IOU=0.5) | | | $Ped.$(IOU=0.25) | | | $Cyc.$(IOU=0.25) | | |
| | | Easy | Mod. | Hard | Easy | Mod. | Hard | Easy | Mod. | Hard |
|---|---|---|---|---|---|---|---|---|---|---|
| PointPillars [8] | L | 63.07 | 54.00 | 54.01 | 38.53 | 37.20 | 37.28 | 38.46 | 22.60 | 22.49 |
| SECOND [23] | L | 71.47 | 53.99 | 54.00 | 55.16 | 52.49 | 52.52 | 54.68 | 31.05 | 31.19 |
| MVXNet [20] | LC | 71.04 | 53.71 | 53.76 | 55.83 | 54.45 | 54.40 | 54.05 | 30.79 | 31.06 |
| ImvoxelNet [19] | C | 44.78 | 37.58 | 37.55 | 6.81 | 6.746 | 6.73 | 21.06 | 13.57 | 13.17 |
| BEVFormer [10] | C | 61.37 | 50.73 | 50.73 | 16.89 | 15.82 | 15.95 | 22.16 | 22.13 | 22.06 |
| BEVDepth [9] | C | 75.50 | 63.58 | 63.67 | 34.95 | 33.42 | 33.27 | 55.67 | 55.47 | 55.34 |
| BEVHeight [24] | C | 77.78 | 65.77 | 65.85 | 41.22 | 39.29 | 39.46 | 60.23 | 60.08 | 60.54 |
| **MonoUNI(Ours)** | C | 90.92 | 87.24 | 87.20 | 51.78 | 49.10 | 48.02 | 69.05 | 69.80 | 69.64 |

M, L, C denotes modality, LiDAR, camera respectively.

Table 3: **Monocular 3D detection performance of Car category on DAIR-V2X-I *val* set.**

# 3 Repartitioning of the Rope3D dataset

To investigate the influence of focal length diversity on model performance, we partitioned the Rope3D [25] dataset according to the focal length of each image. We observed that the focal lengths of the Rope3D dataset predominantly fell within two focal length ranges: 2150-2200 and 2749-2780. Based on the provided training-test distribution from the official website and taking into account the focal length of each image, we reorganized the Rope3D dataset as follows:

Table 4: **The number of images in each subset of the Rope3D dataset.**

| numbers | 2100 | 2700 | all |
|---|---|---|---|
| Train | 20247 | 20086 | 40333 |
| Val | 3149 | 1527 | 4676 |

We conducted training and evaluation on three distinct sets using the widely adopted two-stage approach GUPNet [13], the single-stage approach SMOKE [11], and our proposed method MonoUNI, respectively.

Table 5: **Analysis for different focal lengths on Rope3D dataset with new train/val division.**

| Method | Train_set | $AP_{3D}(IOU = 0.5 \| R_{40})$ | | |
| --- | --- | --- | --- | --- |
| | | val_2100 | val_2700 | val_all |
| GUPNet [13] | train_2100 | 13.20 | 0.03 | 7.42 |
| GUPNet [13] | train_2700 | 0.17 | 21.65 | 3.07 |
| GUPNet [13] | train_all | 10.82 | 5.85 | 9.38 |
| SMOKE [11] | train_2100 | 9.77 | 0.13 | 6.19 |
| SMOKE [11] | train_2700 | 0.04 | 23.20 | 3.64 |
| SMOKE [11] | train_all | 6.04 | 18.01 | 8.48 |
| MonoUNI | train_2100 | 25.78 | 21.30 | 24.47 |
| MonoUNI | train_2700 | 5.77 | 23.42 | 9.25 |
| MonoUNI | train_all | 26.63 | 38.10 | 28.91 |

As shown in Table 5, due to the presence of ambiguity issues arising from focal length and mounting angles, the ordinary vehicle-side 3D detection methods exhibited lower AP on individual focal length testing subsets when trained on the entire dataset, compared to training them separately on the corresponding focal length training subsets. MonoUNI introduces the concept of normalized depth to address the ambiguity issue, effectively mitigating the mutual interference problem between two different focal length training subsets. By employing this approach, MonoUNI achieves state-of-the-art average precision (AP) performance across all testing subsets. Interestingly, when training solely on the $train\_2100$ subset, the model achieved a level of accuracy on the $val\_2700$ subset that was comparable to models trained exclusively on the $train\_2700$ subset. This further highlights the effectiveness of our method.

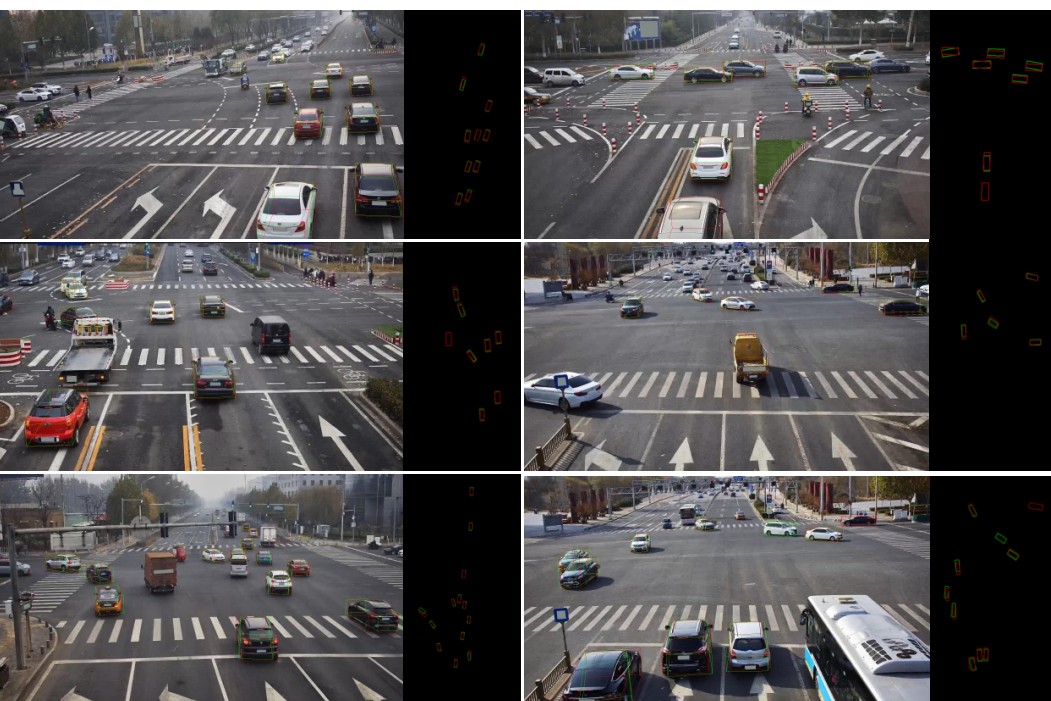

Figure 2: **Qualitative Results on Rope3D.** The 3D green boxes are produced by MonoUNI and the red boxes are the ground truths.

# 4   More Qualitative Results

We provide more qualitative results in Figure 2 and Figure 3.

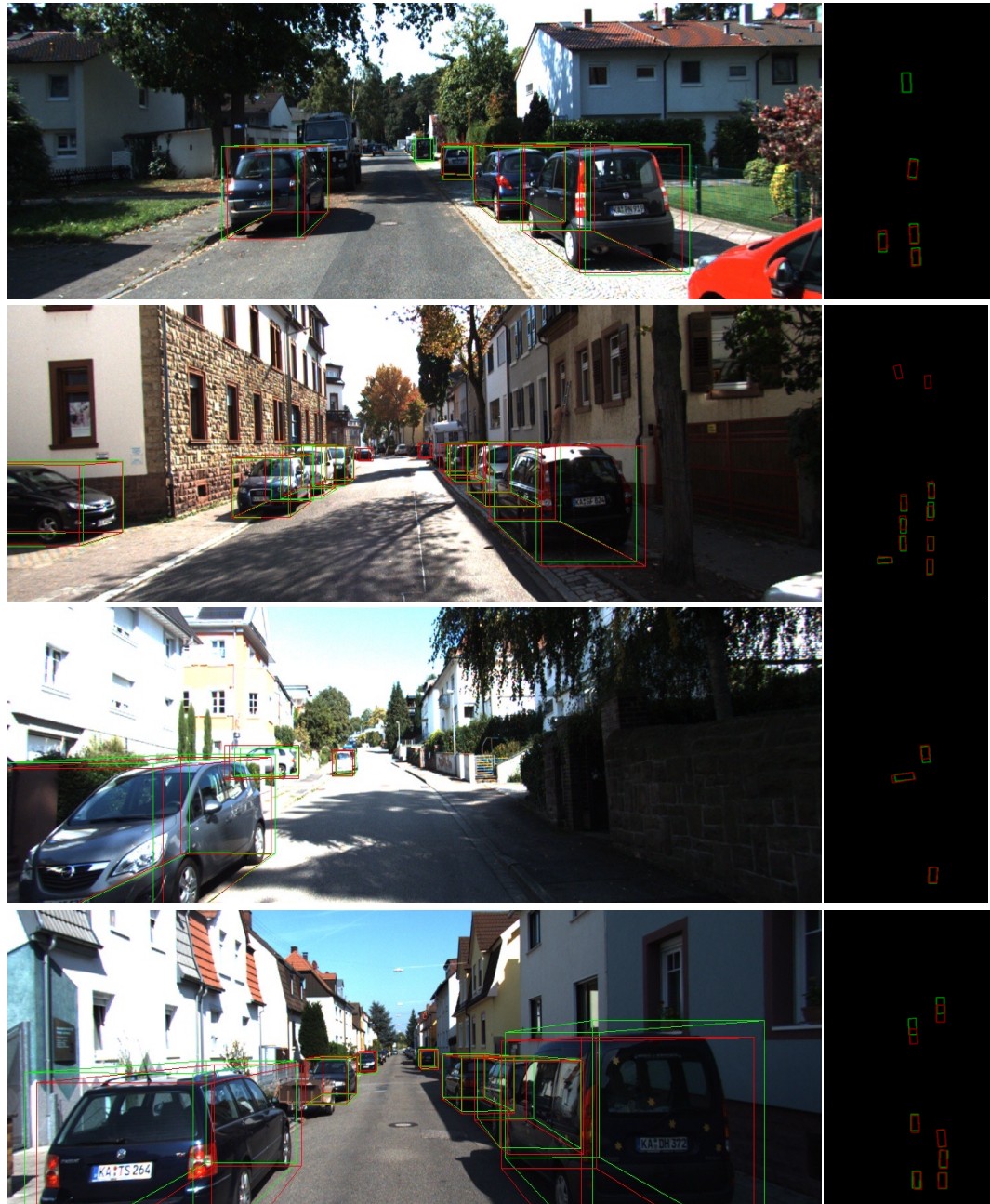

Figure 3: **Qualitative Results on KITTI.** The 3D green boxes are produced by MonoUNI and the red boxes are the ground truths.