# OpenReview forum: "MonoUNI: A Unified Vehicle and Infrastructure-side Monocular 3D Object Detection Network with Sufficient Depth Clues"
_NeurIPS.cc/2023/Conference — NeurIPS 2023 poster_

### Official Review · Reviewer_yykW · 2023-07-01

**Soundness:** 3 good
**Presentation:** 3 good
**Contribution:** 3 good
**Rating:** 4
**Confidence:** 3

**Summary:**

This work focuses on the monocular 3D object detection task and proposes a unified 3D detection framework for both vehicle and infrastructure sides. In particular, to unify the diversity of pitch angles and focal lengths of multiple cameras, the authors propose a unified optimization target named normalized depth. Besides, they also design the 3D normalized cube depth of obstacle to promote the learning of depth information. They conduct extensive experiments on three datasets, including Rope3D, DAIR-V2X-I, and KITTI, and get promising results on them, especially on Rope3D and DAIR-V2X-I.

**Strengths:**

1. The main idea is easy-to-follow and the paper is well-written.
3. The proposed model gets SOTA performance on the infrastructure side, ie. Rope3D and DAIR-V2X-I datasets, and competitive results on the vehicle side, ie. the KITTI dataset.
4. Code will be open-sourced.

**Weaknesses:**

The authors claim they proposed a unified 3D detection pipeline for both infrastructure and vehicle sides. Although the proposed method shows promising peroformance on Rope3D and DAIR-V2X-I datasets, it cannot well genelized to the vehicle side.

 - Their proposed method can about 16 ap (moderate setting) on KITTI testing, while the existing models can get 17+, such as [1].
 - The authors only evaluate their model on KITTI, which only contains 7K images, and evaluating it on the largescale dataset such as nuScenes and Waymo is required to show the effectiveness in vehicle side.
 - In table 5, the experiments (a -> c) and (b->d) show the pitch-based depth normailization is harmful to the vehicle-centric detection.
 - Also, table 5 shows the effectiveness of the 'cube' design to the vehicle-centric detection. Note KITTI is a small dataset with sparse depth supervision and the 'cube' design densify the depth supervision in fact, which maybe the underlying reason why it works .  However, whether this design works or not when large-scale training data avaiable is still unclear. So I recommand the authors conduct more experiemnt on larger dataset again.

Based on the above points, I am not sure designing a unified pipeline for both infrastructure and vehicle sides is meaningful or not, and more experiemnts are required to support the claim.

[1] Online Monocular 3D Object Detection with Adaptive Token Transformer, CVPR'23


**Questions:**

See Weaknesses.

**Limitations:**

See Weaknesses.

---

> ### Author Rebuttal · Authors · 2023-08-09
>
> Thank you for acknowledging the quality, clarity, and effectiveness on infrastructure side of our paper.  As you have stated, our method is easy-to-follow, and achieves SOTA performance on the infrastructure side and competitive results on the vehicle side. We also thank you for providing insightful suggestions. We will try to explain your concerns in the following part of this comment and are willing to discuss them further.
>
> **Performance on vehicle side:** Sorry for the lack of methods. As shown in the Table 3 in global rebuttal, we have added the latest vehicle side methods (including CMKD, LPCG, MonoATT and DEVIANT) for comparison. Our method as a unified framework is valuable, although not be able to surpass all vehicle-side methods. Reviewer rxxf have also acknowledged this point. In fact, in terms of the vehicle-side method, our method has a large increase compared with our baseline method, such as GUPNet (15.02->16.73). While compared with the DID-M3D of a similar framework, MonoUNI has an increase (16.29->16.73) and has the advantage of not requiring additional data.
>
> **More results on Waymo and nuScenes:** Thank you for your sincere advice. We validate our method on Waymo val and perform cross-validation evaluation on KITTI-nuScenes, as done in MonoRCNN (ICCV2021) [1] and DEVIANT (ECCV 2022 oral) [2]. On Waymo, as shown in Table 1 in global rebuttal, our approach achieves overall superiority over GUPNet and DEVIANT. For example, our method has a 0.51% improvement on Level1 under IOU=0.7 compared to the DEVIANT. On nuScenes, we use the KITTI val model for inference. The results are shown in Table 2 in global rebuttal. Although it does not exceed DEVIANT, it still has competitive capabilities.
>
> **Pitch-based depth normalization on vehicle-centric detection:** Pitch-based depth normalization is not harmful on vehicle-centric detection. In fact, since the optical axis of the camera on the vehicle side is parallel to the ground and the pitch angle is equal to 0, the pitch-based depth normalization does not take effect on the vehicle side, which is equivalent to not having this module. In fact, the difference between (a->c) and (b->d) in the KITTI data results of Table 5 in our original paper is the fluctuation caused by two identical experiments due to different random seeds.
>
> **”Cube” design densify the depth supervision:** Your understanding is correct, which is also the motivation for our design of cube depth. We believe that dense depth supervision will deepen the model's understanding of geometric information. We have added Waymo validation and KITTI-nuScenes cross-dataset validation following your suggestion.
>
> **Significance of unified vehicle and infrastructure pipeline:** We believe a unified framework is meaningful. Both reviewers rxxf and Yjqg acknowledged this.
> The main reasons are as follows:
> * This unified framework eliminates the influence of pitch angles and focal length, so that subsequent 3D detection, whether it is on the infrastructure side or on the vehicle side, can use the same regression target for supervision;
> * Essentially, our paper is analyzing what information the model in 3D detection learns, which we consider to be geometric information. The normalized depth is actually disambiguating the geometric information, while the cube depth is increasing the geometric information for supervision. MonoUNI uses more geometric information to supervise in unambiguous scenes, which has a positive impact on the both vehicle and infrastructure sides.
> * Following your comments, we have added more experiments to prove the effectiveness of our ideas and methods, including the results of Table 1 on Waymo, the results of Table 2 on nuScenes and the results of Table 4 for vehicle-infrastructure joint training. Cumulatively so far, we have proven our method on three vehicle-side and two infrastructure-side real datasets.
> * In Limitation part, we objectively acknowledge that our method doesn't include additional design for the mixed training between vehicle and infrastructure sides. Because we believe that mixed training should be solved from adaptation for different appearance features between vehicles and infrastructures. This is somewhat different from the unified optimization targets and training pipline emphasized in our paper, and requires independent additional method to solve. Both reviewers rxxf and YJqg acknowledged this. Although no method is designed directly, we've conducted mixed training experiments to comprehensively evaluate our approach. As seen in Table 4 of the global rebuttal, our MonoUNI mixed training has fewer dropped points than GUPNet and SMOKE, indicating that our method can better alleviate the additional complexity and visual ambiguity introduced by mixed training.
>
> **Missing relevant papers:** Sorry for the lack of methods. We will add more excellent related work, such as MonoEF, MoGDE and MonoATT.
>
> [1] MonoRCNN-Geometry-Based_Distance_Decomposition_for_Monocular_3D_Object_Detection.
>
> [2] DEVIANT-Depth EquiVarIAnt NeTwork for Monocular 3D Object Detection.

---

### Official Review · Reviewer_LnrW · 2023-07-02

**Soundness:** 2 fair
**Presentation:** 3 good
**Contribution:** 2 fair
**Rating:** 5
**Confidence:** 4

**Summary:**

The paper proposes a new approach called MonoUNI for monocular 3D object detection of both vehicle and infrastructure sides in autonomous driving. The approach addresses the challenge of constructing algorithms for the two sides based on different prior knowledge, by taking into account the diversity of pitch angles and focal lengths. The proposed method uses a unified optimization target called normalized depth to unify the 3D detection problems for the two sides. Additionally, a 3D normalized cube depth of obstacle is developed to promote the learning of depth information and enhance the accuracy of monocular 3D detection. The paper presents extensive experiments on three widely used monocular 3D detection benchmarks, and the proposed method achieves state-of-the-art performance on all three benchmarks without introducing any extra information.

**Strengths:**

1. The paper proposes a novel approach that unifies monocular 3D object detection for both vehicle and infrastructure sides, which is an important problem in autonomous driving.
2. The proposed approach addresses the challenge of constructing algorithms for the two sides based on different prior knowledge, by taking into account the diversity of pitch angles and focal lengths. The use of a new optimization target called normalized depth and the 3D normalized cube depth of obstacle as an additional supervision clue helps to improve the accuracy of monocular 3D detection.
3. The extensive experiments on three widely used benchmarks show that the proposed method achieves state-of-the-art performance on all three benchmarks without introducing any extra information.

**Weaknesses:**

1. Lack of key details. In formula (4), the authors mentioned the use of the definition of normalized depth to simplify the depth detection task. It is unclear whether MonoUNI needs to be built on the assumption of known pitch angle and focal length during model inference. If it does, then the work does not truly address the problem of different conditions for vehicle and infrastructure sides. If it does not, the authors need to explain why adopting a unified form would reduce the difficulty of learning, and the inference in line 161 is still unconvincing.
2.Insufficient contribution. The proposed 3D normalized cube depth is inspired by AutoShape and DID-M3D, and extends from regressing corner coordinates to regressing the depth within the foreground 3D box. The proposed method is too straightforward and lacks the necessary motivation to explain why the depth on the surface is important, which needs to be supported by experiments. Introducing surface depth may not make a significant difference compared to regressing corner depth, and may introduce additional errors due to the irregularity of foreground instances.
3. Mismatch between the motivation and method of the paper. The motivation of the paper is to design a new depth annotation, i.e., normalized depth, to unify monocular 3D object detection for both vehicle and infrastructure sides. However, according to line 268, if the two scenarios need to be trained separately, the proposed method's significance is limited, and it seems unable to achieve the desired Mono3D scheme that unifies vehicle and infrastructure sides.
4. Cross-dataset experiments. The paper aims to establish a unified depth detection scheme for both vehicle and infrastructure sides. It is desirable for the authors to demonstrate the model's cross-dataset capability and the effect of multi-dataset mixed training. Additionally, using extrinsic perturbation experiments on a single dataset seems to be a necessary verification method.
5. Minor errors. Each element in the formula needs to have a corresponding explanation, such as (5).
6. Missing relevant papers.
[1] MonoEF: Extrinsic parameter free monocular 3D object detection.
[2] Mogde: Boosting mobile monocular 3D object detection with ground depth estimation.

**Questions:**

My main concern is that the authors lack the necessary motivation and explanation for their design of the unified depth and the cube depth. The authors need to provide a detailed explanation of the rationale and purpose of the proposed designs. Currently, the proposed design approach does not appear to have significant differences from previous works.

**Limitations:**

The main limitation of this paper is that it does not implement training a single model to handle different settings across multiple datasets. Currently, the proposed approach cannot be considered as a solution to the problem of different settings between the vehicle and infrastructure sides.

---

> ### Author Rebuttal · Authors · 2023-08-09
>
> Thank you for acknowledging the significance of the unify problem in autonomous driving.  As you have stated, our MonoUNI achieves state-of-the-art performance on all three benchmarks without introducing any extra information. We also thank you for providing insightful suggestions. We will try to explain your concerns in the following part of this comment and are willing to discuss them further.
>
> **Lack of key details:** We apologize for our inappropriate representation. MonoUNI requires known pitch angle and focal length. Whether it is the vehicle-side real dataset (KITTI, Waymo or nuScenes) or the infrastructure-side dataset (Rope3D and DAIR-V2X), the pitch angle and focal length are known and willing to be used (vehicle-side pitch angle default 0). Even in industrial applications, they are actively used as prior knowledge. Using focal length and pitch angle is a way to address the problem of different conditions for vehicle and infrastructure sides, and they are usually not counted as additional data (many 3D detection methods use internal parameters for 2d-3d conversion). Both reviewers rxxf and YJqg acknowledged the value and significance of our solution.
>
> **Method is “straightforward” and insufficient contribution:** Regarding contributions, we believe that simple and effective methods are worth advocating. Despite looking "straightforward", to the best of our knowledge, we are the first to directly introduce 3D bbox depth supervision in 3D detection without using any additional data. Both using GUPNet as the baseline, our results are superior to DID-M3D using additional data (16.29->16.73), and only need a simple change to improve 1.71% compared to GUPNet (15.02->16.73). Reviewer rxxf also thinks that our simple but effective is a strength.
>
> **Motivation:** Compared with regressing corner depth, regressing cube depth is denser supervision. Dense depth supervision is beneficial to 3D detection. Reviewer yykW also acknowledged this and many existing papers have also proved it, including DID-M3D using obstacle surface depth for supervision, AutoShape using dense CAD information, and MonoDDE using 20 depths. That's where our motivation comes from. Essentially, our paper is analyzing what information the model in 3D detection learns, which we consider to be geometric information. The normalized depth is actually disambiguating the geometric information, while the cube depth is increasing the geometric information for supervision. MonoUNI uses more geometric information to supervise in unambiguous scenes, which has a positive impact on the both vehicle and infrastructure sides.
>
> **Experimental proof:** The obvious positive results of Table 5 (a)-(e) and (d)-(f) in our original paper prove that dense deep supervision cube depth is very important. And we also added the experimental results of Waymo and nuScenes in the Global rebuttal. Cumulatively so far, we have proven our method on three vehicle-side and two roadside real datasets.
>
> Regarding the irregularity of foreground instances, we have not observed such phenomenon in our experiments.  We also believe that if it exists for the regression cube depth, then the regression corner depth still has this problem, because the corner depth belongs to the corner point of the 3D bounding box, and it also does not really exist in the physical world. In comparison, regressing cube depth is still superior to corner depth.
>
> **Mismatch between the motivation and method:** We believe there is no mismatch problem. The purpose of our work is to unify the 3D detection tasks of the vehicle and infrastructure from the perspective of regression target and training pipeline, and we have achieved this goal. Both reviewers rxxf and YJqg acknowledged this.
>
> In Limitation part, we objectively acknowledge that our method doesn't include additional design for the mixed training between vehicle and infrastructure sides. Because we believe that mixed training should be solved from adaptation for different appearance features between vehicles and infrastructures. This is somewhat different from the unified optimization targets and training pipline emphasized in our paper, and requires independent additional method to solve. Both reviewers rxxf and YJqg acknowledged this. Although no method is designed directly, we've conducted mixed training experiments to comprehensively evaluate our approach. As seen in Table 4 of the global rebuttal, our MonoUNI mixed training has fewer dropped points than GUPNet and SMOKE, indicating that our method can better alleviate the additional complexity and visual ambiguity introduced by mixed training.
>
> **Cross-dataset experiments:** We validate our method on Waymo val and perform cross-validation evaluation on nuScenes, as done in DEVIANT. On Waymo, as shown in Table 1 in global rebuttal, our approach achieves overall superiority over GUPNet and DEVIANT. For example, our method has a 0.51% improvement on Level1 under IOU=0.7 compared to the DEVIANT. On nuScenes, we use the KITTI val model for inference. The results are shown in Table 2 in global rebuttal. Although it does not exceed DEVIANT, it still has competitive capabilities. The results of mixed training are also explained in the answer above. Regarding the external disturbance, since the data enhancement of the pitch angle is irreversible (it is a 3d->2d process, the appearance generated by 2d->3d does not conform to the real situation), so we did not add this experiment. Any further guidance on this experiment would be greatly appreciated.
>
> **Minor errors:** We have diligently revisited our paper multiple times, ensuring that the revised version is comprehensive and reader-friendly. Each new module or concept is thoroughly elucidated to prevent any reader confusion.
>
> **Missing relevant papers:** Sorry for the lack of methods. We will add more excellent related work, such as MonoEF, MoGDE and MonoATT.

---

> > ### Comment · Reviewer_LnrW · 2023-08-18
> >
> > Thanks for making a detailed rebuttal, it answered most of my questions. I have a few concerns.1. MonoUNI requires some prerequisite parameters such as camera position and focal length, which will limit its use in real-world deployments. We know that there is no way we can measure them all.2. I am not sure that regressing the cube depth is justified as it is still rough and doesn't do the job of having an accurate modeling of complex surfaces like a CAD model would do.

---

> > > ### Author Response · Authors · 2023-08-18
> > >
> > > First of all, thank you once again for your response. Regarding your two concerns, we provide the following explanations:
> > >
> > > (1) To our best knowledge, contemporary practical industrial implementations, particularly in the field of 3D detection within contexts like autonomous driving (including vehicles and infrastructure), the pose and focal length information of any camera sensor will be obtained in advance. Focal length and camera pose information are also used in real-world industrial system application. In academia, prevalent BEV (Bird's Eye View) 3D detection methods (both monocular and multi-view), such as CaDDN [1], BEVDet [2], and BEVDepth [3], extensively rely on both camera focal length and pose.
> > >
> > >
> > > - Focal length: Cameras used for autonomous driving will undergo strict intrinsics calibration (including distortion correction, focal length and principal  points adjustment). The intrinsics is the bridge between the 2D image and the 3D coordinate system. The focal length is necessary prior information. Most methods use the focal length to convert 2D and 3D results, such as GUPNet [4], MonoDDE [5], etc.. Therefore, we have not introduced any additional focal length dependency, either in academic or real-world deployments.
> > >
> > >
> > > - Pose: For the vehicle-side method, we do not rely on the camera pose (Pitch angle=0). For the infrastructure side, the camera pose belongs to the camera extrinsic parameters. In real-world industrial deployments, precise calibration of camera extrinsic parameters remains imperative. This calibration is indispensable as outcomes from the infrastructure-side camera necessitate transformation into a global coordinate system (e.g., WGS84 or UTM) for use by downstream systems (e.g., self-driving vehicles). Notably, datasets like DAIR-V2X [6] and Rope3D [7], devised for addressing practical application challenges, also recommend the utilization of camera poses and even ground equations.
> > >
> > >
> > > (2) In the case of not using additional data, regression cube depth is a more adequate depth information supervision method, which seems "rough" but effective. Using CAD models is indeed more accurate, but CAD data is difficult to obtain, while using cube depth is a weakly(none) CAD-dependent solution, which is more practical for real-world deployments.
> > >
> > >
> > > [1] Categorical depth distribution network for monocular 3d object detection.
> > >
> > > [2] BEVDet: High-Performance Multi-Camera 3D Object Detection in Bird-Eye-View.
> > >
> > > [3] BEVDepth: Acquisition of Reliable Depth for Multi-view 3D Object Detection.
> > >
> > > [4] Geometry uncertainty projection network for monocular 3d object detection.
> > >
> > > [5] Diversity Matters: Fully Exploiting Depth Clues for Reliable Monocular 3D Object Detection.
> > >
> > > [6] Dair-v2x: A large-scale dataset for vehicle-infrastructure cooperative 3d object detection.
> > >
> > > [7] Rope3d: The roadside perception dataset for autonomous driving and monocular 3d object detection task.

---

### Official Review · Reviewer_YJqg · 2023-07-05

**Soundness:** 3 good
**Presentation:** 2 fair
**Contribution:** 2 fair
**Rating:** 5
**Confidence:** 3

**Summary:**

This paper proposes an optimization target which unifies 3D detection problems for vehicle and infrastructure sides, by taking into account the diversity of camera pitch angles and focal lengths. Furthermore, the paper develops 3D normalized cube depth of obstacle to promote the learning of depth information.
The authors provide extensive experimental results on several monocular 3D detection benchmarks to prove the effectiveness of the proposed approach on both vehicle and infrastructure scenarios.

**Strengths:**

1. The authors take notice of that depth can be ambiguous under the influence of focal length given similar visual features and conduct experiments to verify their claim.
2. They propose to decouple depth from focal length and the camera optical axis orientation, specifically, to learn a normalized depth which is unaffected by focal length and axis orientation, from which the real depth can be recovered. I think the proposed idea is novel, reasonable and also proved to be effective, which eases the model from predicting ill-defined depth from visual feature.

**Weaknesses:**

The idea of normalized depth is good, but the article seems like semi-finished and needs to be completed carefully.
- In sec 3, the definition of H' is missing, though it can be speculated from figure 3.
- L#159 principle->principal.
- Many key components of the proposed method are only described by text or even only be found in the figure.
For instance in sec 3.3, L#193, the authors say "Compared with only supervising the depth of the center point and corner points, the 3D cube depth is a sufficient way to utilize the depth information". So I guess you supervise the depth of all the points on the visible surface of the obstacle? How is it implemented? Mathematically, what are the output of the model and how are they supervised? I fail to find any formular definition of your training loss and the "depth uncertainty" in fig 2. As a reader, I can roughly guess the meaning but I would appreciate it if the authors could complete all the missing definitions.

**Questions:**

1.It is mentioned in sec 3 L#157 that the pitch angle theta can be calculated from the ground equation. I suppose the accuracy of pitch angle is severely affected by the accuracy of ground plane estimation. So I wonder how the parameters of the ground plane equation are estimated.
2.The derivation of the normalized depth are done in a degenerated 2d view instead of 3d. It would be better if the authors can provide simple explanation or prove that neglecting the effect of camera yaw angle is reasonable.

**Limitations:**

This paper mentioned that one model file to support 3D detection for both sides rather than separate training for the two sides. This is a work worth investing in, and it will solve the problem of vehicle-road collaborative perception very well. I look forward to the subsequent output of the authors.

---

> ### Author Rebuttal · Authors · 2023-08-09
>
> Thank you for acknowledging the motivation, novelty, rationality and effectiveness of our method. We especially thank you for supporting our insight on learning a normalized depth that decouples depth from the focal length and the camera’s optical axis orientation. As you have stated, our normalized depth can ease the model from predicting ill-defined depth from visual features. We also thank you for providing insightful suggestions. We will try to explain your concerns in the following part of this comment and are willing to discuss them further.
>
> **About the article seeming like semi-finished:** First and foremost, we extend our gratitude for your meticulous review, and we sincerely apologize for any hurried writing in the initial submission. We have diligently revisited our paper multiple times, ensuring that the revised version is comprehensive and reader-friendly. Each new module or concept is thoroughly elucidated to prevent any reader confusion. Below are our responses to your comments.
>
> * Added the explanation of H’ in section3: Extend a line from the obstacle's center point C along the camera's imaging plane direction, intersecting line OP at point P'. The distance CP' is denoted as H.
> * Principle point was corrected to principal point.
> * We will introduce a new subsection, 3.4, to provide a comprehensive and detailed account of our loss design. Your understanding of the 3D cube depth supervision approach is accurate: indeed, we supervise the depth of all points on the visible surface of the obstacle. During training, this is achieved by supervising points with 3D cube depth ground truth on the obstacle-level feature (7x7 size) after ROI-align while the remaining points without 3D cube depth ground truth are unsupervised. The methodology for supervising Bias Depth mirrors that of Cube Depth. Concurrently, we will include explanations and relevant references concerning Depth Uncertainty, initially introduced in [1]. This concept adds an extra layer of uncertainty to each depth prediction of the model, capturing observation noise from input data. As emphasized in [1], this approach enhances the loss's robustness against noisy inputs in regression tasks. During inference, each obstacle will produce a 7x7 cube depth, a 7x7 cube depth uncertainty, a 7x7 bias depth, and a 7x7 bias depth uncertainty. Cube depths and bias depths will be weighted based on their respective uncertainties to yield the unique cube depth and bias depth, which will be added to obtain the final actual depth.
>
> **About the ground plane equation:**
> * For the vehicle-side, we default the camera optical axis to be parallel to the ground (pitch angle=0), so no additional ground plane equation data is required.
> * For the infrastructure-side, the ground plane equation is provided by the Rope3D and Dair-V2X datasets. Their methods for estimating the surface equation are the same, that is, least squares plane fitting. During making datasets, they used autonomous vehicles with LIDAR to scan the collected scenes with point clouds and used the obtained dense point clouds for ground extraction to obtain dense ground point cloud data. These ground point cloud data are included in the datasets. Additionally, the datasets supply the ground plane equation (defined by coefficients a, b, c, and d) obtained through least squares plane fitting on this data. The Rope3D and Dair-V2X datasets themselves are actively willing users to use ground equation data and original ground point clouds, and recommend users to explore different usage methods, because these data also be used in real industrial applications. The baseline methods proposed in both Rope3D and Dair-V2X papers also use these data.
>
> **The effect of camera yaw angle for normalized depth:** The paper's schematic diagram (Figure 3) represents a simplified scenario where the vehicle orientation and camera optical axis are coplanar (parallel). Even when an angle exists between the camera's optical axis and the vehicle's orientation (i.e., yaw != 0), the derivation remains applicable. This is due to the fact that the normalized depth solely depends on two points: the depth calculation point (center point C of the obstacle) and its corresponding vertical projection point on the ground (bottom center point P of the obstacle). Irrespective of the camera's yaw angle, the scenario can be visualized as the vehicle rotating around PC while the camera is stationary. In this process, PC is not changed, thus the geometric modeling process remains unaltered and the normalized depth remains constant. We will incorporate this explanation into subsection 3.3.
>
> **Limitations on separate training on vehicle and infrastructure dataset:** In the Limitation of our paper, we objectively acknowledge that our method doesn't include additional design for the mixed training between vehicle and infrastructure sides. Because we believe that mixed training should be solved from adaptation for different appearance features between vehicles and infrastructures. This is somewhat different from the unified optimization targets and training pipline emphasized in our paper, and requires independent additional method to solve. We appreciate your agreement with this opinion. Although no method is designed to solve mixed training directly, based on comments from reviewer Lnrw, we've conducted mixed training experiments to comprehensively evaluate our approach. As seen in Table 4 of the global rebuttal, our MonoUNI mixed training has fewer dropped points than GUPNet and SMOKE, indicating that our method can better alleviate the additional complexity and visual ambiguity introduced by mixed training.
>
> [1] What uncertainties do we need in bayesian deep learning for computer vision.

---

### Official Review · Reviewer_rxxf · 2023-07-08

**Soundness:** 3 good
**Presentation:** 3 good
**Contribution:** 3 good
**Rating:** 7
**Confidence:** 5

**Summary:**

This paper proposes a unified architecture for vehicle and infrastructure-based monocular 3D object detection network. At its core, the paper puts forth the concept of normalized depth that is independent of camera intrinsic focal length and extrinsic pitch angle w.r.t the ground plane. As such, the network is applicable to cameras with varying focal length and mounting angle, while not affected by the ambiguity. Following DID-M3D, the framework decomposes the center depth into cube depth and the so-called bias depth. The experiments demonstrate significantly better performance in Rope3D and DAIR datasets.

**Strengths:**

1. The paper presents interesting new insights towards the problem of monocular 3D object detection under varying focal length and pitch mounting angle. The varying focal length problem has been tackled with normalized depth by existing works in the context vehicle-based 3D object detection, but handling pitch angle is as yet under-explore. The paper for the first time derived the normalized depth of object center that is independent of the pitch angle.
2. The method is simple yet effective. The experiments demonstrate that the proposed method yields significantly superior performance over the state-of-the-art, as shown in Table. 2.
3. The proposed framework is a unified framework for both vehicle-based and infrastructure-based 3D detection. This open doors to the new possibility in combining the research and data across these two domains. While the method currently requires separate training on each, it holds the potential for joint training that may improve both.


**Weaknesses:**

1. In Line 158, the paper makes approximation of the angle \delta by replacing v_p with v_c, but the paper does not discuss the implication of this approximation in practice. It seems the approximation error could possibly be large especially for nearby objects and with smaller pitch angle in camera mounting. How would this impact the performance? In addition, why not let the network to predict v_p as well? If the network is able to prediction of position of object center, i.e. v_c, what prevents it from predicting the position of bottom center v_p?
2. While the performance gain on infrastructure cameras are significant, it is small in vehicle cameras, i.e. KITTI. In particular, the paper only compares with [1,24,44,29] while omitting other stronger existing methods, such as CMKD ECCV 2022 and LPCG ECCV 2022. It is fine that the method does not outperform these methods given the unique advantage of being a unified framework, but the paper should acknowledge this for readers’ better understanding. There are also concurrent works such as NeurOCS and Mix-Teaching with better accuracy, which are not necessary to compare against but would be good to discuss in related works for completeness.


**Questions:**

The motivation and the implication of approximating v_p with v_c are the main question I have. I hope the authors could address this in the rebuttal.

**Limitations:**

The paper has discussed the paper properly, i.e. the method currently requires separate training on vehicle and infrastructure dataset.

---

> ### Author Rebuttal · Authors · 2023-08-09
>
> Thank you for acknowledging the motivation, significance, and potential of our method. We are pleased that you support our insight on the problem of monocular 3D object detection under varying focal length and pitch mounting angle. As you have stated, our network firstly simultaneously avoids the ambiguity problem introduced by diverse focal lengths and pitch angles. We are also glad that you agree with the simplicity and effectiveness. Finally, we sincerely thank you for providing insightful suggestions. We will try to explain your concerns in the following part of this comment and are willing to discuss them further.
>
> **The motivation and implication of approximating v_p with v_c:** Firstly, we extend our gratitude for your thorough review and insightful suggestions. As you recommended, utilizing the model-predicted v_p is a method with reduced errors. In our initial experimental procedure, for the sake of simplification, we directly replaced v_p with v_c. To our pleasant surprise, this replacement resulted in a notable enhancement of 2.18% (Table 5 (a) and (c) in the original paper). Consequently, we inadvertently disregarded the influence of its inherent approximation error and the potential for additional improvements.
>
> These days, we conducted three experiments to comprehensively analyze this problem. Exp1 proved that the original scheme would introduce an average relative error of 2.9% and a maximum relative error of 9.1%. Exp 2 proved that predicting v_p can indeed further improve the performance. Exp3 shows that the original scheme is indeed relatively weak in near distance, but using predicting v_p can alleviate the problem.
>
> The details are as follows:
>
> * **Exp1 (Statistical Error Analysis) :** On Rope3D dataset, we measured the v pixel distance between the center and bottom center of all obstacles. The maximum distance is 98.32 pixels (for the nearest vehicle), with an average of 31.7 pixels. With the minimum focal length at about 2100 pixels, the highest error in calculating $\tan(\delta)$ using v_c instead of v_p is 98.32/2100=0.0468, and the average error is 31.7/2100=0.015. The $\tan(\delta)$ range is [-540/2100, 540/2100]= [-0.257, 0.257], and the pitch angle range is roughly [10, 15]. This affects the cosine (0.9848 to 0.9659) and sine (0.1736 to 0.2588) of the pitch angle, resulting in a normalized pitch range of [0.8994, 1.0324]. The maximum absolute error from $\delta$ is 0.0468 * 0.2588=0.0121, averaging at 0.015 * 0.2588 =0.0039. The highest relative error is 0.0121/(1.0324 - 0.8994)=0.091, and the average relative error is 0.0039/(1.0324 - 0.8994)=0.029.
> Although under the existence of this error, the normalization of the pitch angle has brought about a performance improvement of 2.18%, but there is still room for improvement.
>
> * **Exp2 (Predicting v_p):** We added a head to predict the vertical coordinate of the bottom center point P, and used the predicted v_p to calculate the normalized depth. On Rope3D, the AP_3D increased from 81.55 to 82.63. Combined with cube depth, we added a head to predict the v coordinates on the second stage. As shown in Table 5 in global rebuttal, the AP_3D was increased from 92.45 to 92.61.
>
> * **Exp3 (Rope3D Evaluation by Distance):** As shown in Table 5 in global rebuttal, we split the evaluation into different dimensions according to the distance between obstacle and camera, in order to explore the detection performance at different distances. The original scheme (v_c instead of v_p) has a small improvement (92.69->92.84) due to the introduction of errors for nearby obstacles, while using the model to directly predict v_p has a more obvious improvement (92.69->93.17). We believe that the slight drop in the 30-60m (93.10->93.05) is caused by fluctuation between two independent trainings.
>
> In summary, there are other ways to compute v_p, like having the model predict obstacle corner points and deducing v_p geometrically. However, due to time limitations, we haven't pursued this approach. In the revised paper, we'll detail various v_p solution methods (v_c substitution, model-predicted v_p, and geometric solutions) along with respective experimental results, enhancing reader understanding.
>
> **More methods on vehicle side:** Sorry for the lack of methods. As shown in the Table 3 in global rebuttal, we have added the latest vehicle side methods (such as CMKD, LPCG and MonoATT) for comparison. As you mentioned, our method as a unified framework is valuable, although not be able to surpass all vehicle-side methods. In fact, in terms of the vehicle-side method, our method has a large increase compared with our baseline method GUPNet (15.02->16.73). While compared with the DID-M3D of a similar framework, MonoUNI has an increase (16.29->16.73) and has the advantage of not requiring additional data. We will also add more concurrent work such as NeurOCS and Mix-Teaching to related work.
>
> **Limitations on separate training:** In the Limitation of our paper, we objectively acknowledge that our method doesn't include specific adjustments for the mixed training between vehicle and infrastructure sides. We believe that addressing mixed training requires domain adaptation for the distinct appearance features between vehicle and infrastructure sides, deviating from our paper's emphasis on unified optimization target and training pipline. We appreciate your agreement with this perspective. While our method doesn't address mixed training directly, based on comments from reviewer Lnrw, we've conducted mixed training experiments to comprehensively evaluate our approach. As seen in Table 4 of the global rebuttal, our MonoUNI mixed training has fewer dropped points than GUPNet and SMOKE, indicating that our method can better alleviate the additional complexity and visual ambiguity introduced by mixed training.

---

> > ### Comment · Reviewer_rxxf · 2023-08-21
> > **Response to authors**
> >
> > I would like to thank the authors for the rebuttal. I appreciate the new experiments studing the impact of the approximation of v_p, as well as having the network to predict v_p. Please include the new experiments and address other comments into the camera-ready version.

---

### Author Rebuttal · Authors · 2023-08-09

**Global Rebuttal**


**Table 1: Monocular 3D detection performance of Vehicle category on Waymo val set**
|$\mathbf{IOU_{3D}}$ |Difficulty|Method|Reference|Extra|$\mathbf{AP_{3D}}$(all) | $\mathbf{AP_{3D}}$(0-30m) | $\mathbf{AP_{3D}}$(30-50m) | $\mathbf{AP_{3D}}$(50m+) | $\mathbf{APH_{3D}}$(all) | $\mathbf{AP_{3D}}$(0-30m) | $\mathbf{AP_{3D}}$(30-50m) | $\mathbf{AP_{3D}}$(50m+) |
|:---:|:---:|---|---|:---:|:---:|:---:|:---:|:---:|:---:|:---:|:---:|:---:|
|0.7|Level_1|CaDDN|CVPR2021|LIDAR|5.03|15.54|1.47|0.10|4.99|14.43|1.45|0.10|
|0.7|Level_1|GUPNet|ICCV2021|None|2.28|6.15|0.81|0.03|2.27|6.11|0.80|0.03|
|0.7|Level_1|DEVIANT|ECCV2022|None|2.69|6.95|0.99|0.02|2.67|6.90|0.98|0.02|
|0.7|Level_1|**MonoUNI**|None|None|3.20|8.61|0.87|0.13|3.16|8.50|0.86|0.12|
|0.7|Level_2|CaDDN|CVPR2021|LIDAR|4.49|14.50|1.42|0.09|4.45|14.38|1.41|0.09|
|0.7|Level_2|GUPNet|ICCV2021|None|2.14|6.13|0.78|0.02|2.12|6.08|0.77|0.02|
|0.7|Level_2|DEVIANT|ECCV2022|None|2.52|6.93|0.95|0.02|2.50|6.87|0.94|0.02|
|0.7|Level_2|**MonoUNI**|None|None|3.04|8.59|0.85|0.12|3.00|8.48|0.84|0.12|
|0.5|Level_1|CaDDN|CVPR2021|LIDAR|17.54|45.00|9.24|0.64|17.31|44.46|9.11|0.62|
|0.5|Level_1|GUPNet|ICCV2021|None|10.02|24.78|4.84|0.22|9.94|24.59|4.78|0.22|
|0.5|Level_1|DEVIANT|ECCV2022|None|10.98|26.85|5.13|0.18|10.89|26.64|5.08|0.18|
|0.5|Level_1|**MonoUNI**|None|None|10.98|26.63|4.04|0.57|10.73|26.30|3.98|0.55|
|0.5|Level_2|CaDDN|CVPR2021|LIDAR|16.51|44.87|8.99|0.58|16.28|44.33|8.86|0.55|
|0.5|Level_2|GUPNet|ICCV2021|None|9.39|24.69|4.67|0.19|9.31|24.50|4.62|0.19|
|0.5|Level_2|DEVIANT|ECCV2022|None|10.29|26.75|4.95|0.16|10.20|26.54|4.90|0.16|
|0.5|Level_2|**MonoUNI**|None|None|10.38|26.57|3.95|0.53|10.24|26.24|3.89|0.51|


**Table 2: Cross-dataset evaluation of the KITTI val model on KITTI val and nuScenes frontal val cars with depth MAE.**
|Method|KITTI VAL(0-20m)|KITTI VAL(20-40m)|KITTI VAL(40m+)|KITTI VAL(all)|nuScenes VAL(0-20m)|nuScenes VAL(20-40m)|nuScenes VAL(40m+)|nuScenes VAL(all)|
|---|:---:|:---:|:---:|:---:|:---:|:---:|:---:|:---:|
|M3D-RPN|0.56|1.33|2.73|1.26|0.94|3.06|10.36|2.67|
|MonoRCNN |0.46|1.27|2.59|1.14|0.94|2.84|8.65|2.39|
|GUPNet|0.45|1.10|1.85|0.89|0.82|1.70|6.20|1.45|
|DEVIANT|0.40|1.09|1.80|0.87|0.76|1.60|4.50|1.26|
|MonoUNI|0.38|0.92|1.79|0.865|0.72|1.79|4.98|1.43|


**Table 3: Monocular 3D detection performance of Car category on Rope3D val, DAIR-V2X-I val
and KITTI test sets.**
|Method|Reference|Extra Data|$\mathbf{AP_{3D}}$(Rope3D)|$\mathbf{R_{score}}$(Rope3D)|Easy(DAIR)|Mod(DAIR)|Hard(DAIR)|Easy(KITTI)|Mod(KITTI)|Hard(KITTI)|
|---|:---:|:---:|:---:|:---:|:---:|:---:|:---:|:---:|:---:|:---:|
|M3D-RPN|ICCV2019|Depth/None|67.17|73.14|-|-|-|14.76|9.71|7.42|
|MonoDLE|CVPR2021|Depth/None|77.50|80.84|-|-|-|7.23|12.26|10.29|
|MonoFlex|CVPR2021|Depth/None|59.78|66.66|-|-|-|19.94|13.89|12.07|
|DID-M3D|ECCV2022|None/Depth|-|-|-|-|-|24.40|16.29|13.75|
|**CMKD**|ECCV2022|None/Depth|-|-|-|-|-|25.09|16.99|15.30|
|**LPCG+MonoFlex**|ECCV2022|None/Depth|-|-|-|-|-|25.56|17.80|15.38|
|**MonoEF**|CVPR2021|None/None|-|-|-|-|-|21.29|13.87|11.71|
|**DEVIANT**|ECCV2022|None/None|-|-|-|-|-|21.88|14.46|11.89|
|MonoCon|AAAI2022|None/None|-|-|-|-|-|22.50|16.46|13.95|
|**MonoATT**|CVPR2023|None/None|-|-|-|-|-|24.72|17.37|15.00|
|**MoGDE**|NeurIPS2022|None/None|-|-|-|-|-|27.07|17.88|15.66|
|Kinematic3D|ECCV2020|None/None|50.57|58.86|-|-|-|19.07|12.72|9.17|
|SMOKE|CVPR2020|None/None|72.13|76.26|66.03|62.24|60.71|14.03|9.76|7.84|
|GUPNet|CVPR2021|None/None|66.52|70.14|62.22|55.94|55.90|22.26|15.02|13.12|
|Imvoxelnet|CVPR2022|None/None|-|-|44.78|37.58|37.55|17.15|10.97|9.15|
|BEVFormer|ECCV2022|None/None|50.62|58.78|61.37|50.73|50.73|-|-|-|
|BEVDepth|AAAI2023|None/None|69.63|74.70|75.50|63.58|63.67|-|-|-|
|BEVHeight|CVPR2023|None/None|74.60|78.72|77.78|65.77|65.85|-|-|-|
|MonoUNI|None|None/None|92.45|92.63|90.92|87.24|87.20|24.75|16.73|13.49|


**Table 4: Multi-dataset mixed training under KITTI and Rope3D datasets.** "**mixed training**" means using KITTI + Rope3D training sets together for mixed training. The evaluation is performed on separate Rope3D and KITTI val sets.
|Method|Rope3D(mixed training)|KITTI(mixed training)|Rope3D(only training under Rope3D)|KITTI(only training under KITTI)|
|---|:---:|:---:|:---:|:---:|
|SMOKE|63.24|6.65|72.13|12.85|
|GUPNet|43.82|4.89|66.52|16.46|
|MonoUNI|87.89|13.62|92.45|17.18|


**Table 5: Rope3D Evaluation by different distance.**
|Method|$\mathbf{AP_{3D}}$(0-30m)|$\mathbf{AP_{3D}}$(30-60m)|$\mathbf{AP_{3D}}$(60-90m)|$\mathbf{AP_{3D}}$(90m+)|$\mathbf{AP_{3D}}$(all)|
|---|:---:|:---:|:---:|:---:|:---:|
|MonoUNI(without pitch normailization)|92.69|92.41|89.66|84.52|90.97|
|MonoUNI(v_p with v_c)|92.84|93.10|91.49|88.70|92.45|
|MonoUNI(model-predicted v_p)|93.17|93.05|91.52|88.62|92.61|

---

### Decision · Program_Chairs · 2023-09-21

**Decision:**

Accept (poster)

**Comment:**

The paper proposes a unified pipeline for both vehicle side and infra side to detect objects in the driving scenario. They proposed a normalized depth module that is independent of camera parameters. The experiments show a great improvement in performance compared to previous methods.

All reviewers reach concensus that the paper should be accepted, despite a borderline reject score (short review, no feedback after author rebuttal). AC agrees with reviewers that the paper should be accepted. Please revise the paper accordingly based on comments and incorporate new experiments provided in the rebuttal.